# Tungsten Ores of the Dzhida W-Mo Ore Field (Southwestern Transbaikalia, Russia): Mineral Composition and Physical-Chemical Conditions of Formation

**Ludmila B. Damdinova \* and Bulat B. Damdinov**

Geological Institute, Siberian Branch of the Russian Academy of Sciences, 670047 Ulan-Ude, Russia; damdinov@mail.ru

\* Correspondence: ludamdinova@mail.ru; Tel.: +7-9969365077

**Abstract:** This article discusses the peculiarities of mineral composition and a fluid inclusions (FIs further in the text) study of the Kholtoson W and Inkur W deposits located within the Dzhida W-Mo ore field (Southwestern Transbaikalia, Russia). The Mo mineralization spatially coincides with the apical part of the Pervomaisky stock (Pervomaisky deposit), and the W mineralization forms numerous quartz veins in the western part of the ore field (Kholtoson vein deposit) and the stockwork in the central part (Inkur stockwork deposit). The ore mineral composition is similar at both deposits. Quartz is the main gangue mineral; there are also present muscovite, K-feldspar, and carbonates. The main ore mineral of both deposits is hubnerite. In addition to hubnerite, at both deposits, more than 20 mineral species were identified; they include sulfides (pyrite, chalcopyrite, galena, sphalerite, bornite, etc.), sulfosalts (tetrahedrite, aikinite, stannite, etc.), oxides (scheelite, cassiterite), and tellurides (hessite). The results of mineralogical and fluid inclusions studies allowed us to conclude that the Inkur W and the Kholtoson W deposits were formed by the same hydrothermal fluids, related to the same ore-forming system. For both deposits, the fluid inclusion homogenization temperatures varied within the range ~195–344 °C. The presence of cogenetic liquid- and vapor-dominated inclusions in the quartz from the ores of the Kholtoson deposit allowed us to estimate the true temperature range of mineral formation as 413–350 °C. Ore deposition occurred under similar physical-chemical conditions, differing only in pressures of mineral formation. The main factors of hubnerite deposition from hydrothermal fluids were decreases in temperature.

**Keywords:** Dzhida W-Mo ore field; Inkur and Kholtoson deposits; tungsten; fluid inclusions; ore-forming fluids





## 1. Introduction

Tungsten mineralization worldwide is spatiotemporally associated with granitic plutons and forms a large variety of ore deposits including breccias, veins-stockworks, greisens, skarns, pegmatites, and porphyries [1,2]. Despite their being well-described in the literature and having been researched on this quartz vein-type wolframite deposit, there is still considerable uncertainty about key issues related to the genesis of these deposits, such as the source of the ore metal, the origin of the ore fluids, and the mechanisms of ore precipitation.

In Southwestern Transbaikalia (Russia), there are known W-Mo deposits of the Dzhida ore field, characterized by high concentrations of mineralization over a relatively small area. The ore field includes large industrial deposits: Pervomaiskoe, Inkur, and Kholtoson. These are the largest sources of W and Mo in Russia. A number of other economic components (Be, Cd, Pb, Zn, Au, etc.) are also present in the ores. The previous studies showed that all three deposits of the Dzhida ore field are genetically related to the one granitoid intrusion. It is believed that the Mo stockwork was formed first (Pervomaiskoe deposit), the veinlets

with Mo-Be mineralization later, and then the stockwork of hubnerite-containing veinlets (Inkur deposit), the latest of which are the quartz-hubnerite veins of the Kholtoson deposit.

Earlier, we conducted a fluid inclusion study of the Pervomaisky Mo deposit, which is considered to be the earliest in the Dzhida ore field [3]. We also specified the isotopic age of the Mo mineralization and determined metal concentrations in the ore-forming solutions. However, the world literature contains no modern data on mineral composition and formation conditions of the W mineralization of the Dzhida ore field. This is especially true for gas-salt composition of the hydrothermal fluids, P-T parameters of the ore deposition, and metal content in the solutions producing the vein-stockwork W-mineralization. The W ore mineralogy was studied in the second half of the 20th century, but due to lack of precision microanalysis methods at that time, not all mineral phases were reliably diagnosed. In general, for greisen Mo-W (Be) deposits, physical-chemical conditions of formation have not yet been studied in sufficient detail to identify the differences in the physical and chemical parameters of ore-forming fluids in Mo-W deposits.

This research focuses on the study of W mineralization in the Dzhida ore field based on the example of the stockwork Inkur and the Kholtoson vein deposit. More than 90 thousand tonnes of $WO_3$ have been mined from both deposits. By resources and content of $WO_3$ in the ores, the Inkur deposit is comparable to the largest deposits in the world: Hemerdon (Great Britain), Pine Creek (USA), Öndör Tsagaan (Mongolia), Xihuashan deposit (China), and other deposits in China, Kalguta (Altay, Russia), Akchatau (Kazakhstan), etc. [4–8]. The purpose of the study was to clarify the mineral composition of the ores and determine the conditions of formation and composition of solutions producing W mineralization in the Dzhida ore field.

## 2. Regional Geology

The Dzhida ore field is an example of multimetal Mo-W mineralization. Here on a relatively small area, there are three large deposits: the Pervomaisky Mo deposit, and Inkur and Kholtoson W deposits. The ore field geology has been studied by different researchers: M.V. Besova [9], P. I. Neletov et al. [10], E. N. Smolyansky [11], V. I. Ignatovich [12,13], E. P. Malinovsky [14], D. O. Ontoev [15,16], E. E. Baturina, G. S. Ripp [17], I. V. Gordienko et al. [18], A. N. Distanova [19], P. Y. Khodanovich and O. K. Smirnova [20], P. Y. Khodanovich [21], I. V. Chernyshev et al. [22], F. G. Reyf, E. D. Bazheev [23], F. G. Reyf, [24], Povilaitis M. M. [25,26], Stelmachonok K. Z. [27,28], etc.

The Dzhida ore field is located in the southwest of Western Transbaikalia. The region represents the southwestern part of the Sayan-Baikal Fold Belt and has complicated cover-fold geological structure. It is characterized by Late Paleozoic faults with a significant shear component. The Khokhyurta sedimentary-effusive suite, the Modonkul diorite massif, and the multiple-phase Gudzhir granitoid intrusion are part of the geological structure of the Dzhida ore field.

The Dzhida deep fault with an S-oriented strike passes through the central part of the ore field. It is traced by a mélange and blastomilonitization zone with total thickness of 600–1000 m, containing small lenticular steeply falling bodies of altered ultrabasites of the Tsakir series. The Khokhyurta suite occupies the eastern part of the ore field and is composed of metamorphosed sandstones, schists, and limestones, as well as effusives with basic and intermediate composition.

The Modonkul intrusive is a fragment of a large massif, elongated in the northwest direction, composed mainly of quartz diorites [20]. Near the contact of Paleozoic quartz diorites of the Modonkul massif with effusive sedimentary sequences, there is the Gudzhir intrusion producing the Mo-W mineralization. On the modern surface, the intrusion is represented by numerous dikes with acidic composition and the Pervomaisky granite-porphyry stock (Figure 1). Among the rocks of the dike complex, there are observed dikes of gray syenite, gray quartz syenite-porphyry, kersantite, bostonite, and porphyry granite as well.

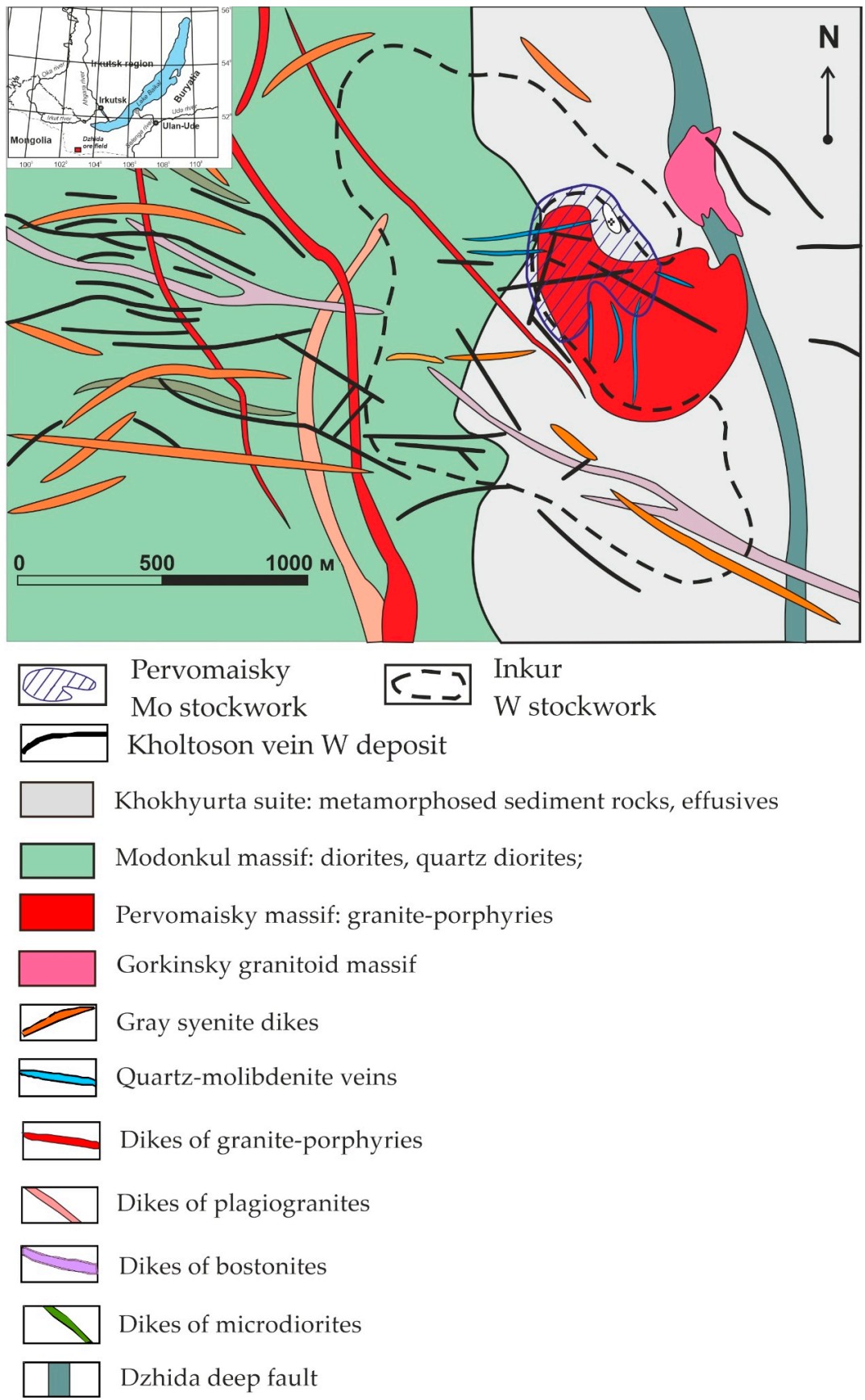

**Figure 1.** Simplified geological map of the Dzhida ore field (according to [21]).

The Pervomaisky granite-porphyry stock is the largest outcrop of the Gudzhir intrusion to the surface. The area of the Pervomaisky massif outcrop is 0.35 km$^2$. The Pervomaisky granite massif is associated with the Mo deposit of the same name. According to drilling and geophysical data, the Pervomaisky massif is a laccolith-shaped body with numerous apophyses; it is elongated in the northwest direction and sinks to the southeast according to the roof of the Modonkul massif. In its apical part, there are xenoliths of granites and syenogranites, as well as roof pendants, indicating the shallow depth of the erosional level of the massif.

The Mo mineralization spatially coincides with the apical part of the Pervomaisky stock, the W mineralization, with numerous quartz veins in the western part of the ore field (Kholtoson) and with the stockwork of the central part (Inkur stockwork) (Figure 1). Recent isotope-geochronological data on the age of zircons (124 ± 2 Ma) from granite-porphyry of the Pervomaisky stock, muscovite (127.6 ± 1.5 Ma), and molybdenite (118.5 ± 1.6 and 122.4 ± 1 Ma) from the ore zones of the Pervomaisky Mo deposit indicate genetic relationships between the processes of granite formation and Mo ore deposition about 119–128 million years ago [3].

## 3. Sampling and Analytical Methods

Ores and wall-rocks were sampled from the eastern and central parts of the Inkur deposit quarry. The Kholtoson deposit rocks were sampled both from the quarry and the underground mine dump. Additionally, some samples from both deposits available at the Geological Institute SB RAS were used. As a result, a representative set of more than 200 samples was collected. The samples were prepared into thin and polished sections to conduct optical microscopic observations, fluid inclusion studies, and SEM EDS analyses.

Mineralogical and petrographic investigations were carried out using the OLYMPUS BX-51 polarization microscope with a MicroPublisher 3.3 RTV digital camera. Chemical composition of the minerals was determined by E.V. Khodyreva and S.V. Kanakin using a LEO-1430VP scanning electron microscope with an energy-dispersive INCA Energy 350 spectrometer at the "Analytical center of mineralogical, geochemical and isotope studies" at the Geological Institute of the Siberian Branch of the Russian Academy of Sciences (Ulan-Ude, Russia).

Fluid inclusions in quartz, fluorite, and hubnerite were investigated by the methods of microthermometry and Raman spectroscopy. To determine the temperature of homogenization, eutectic temperature, and ice melting temperature of aqueous solutions, temperatures of dissolution of daughter phases, a THMSG-600 Linkam stage with a temperature measuring range of −196 to +600 °C was used. The standard instrumental measurement error is ±0.1 in the negative and ±5 °C in the positive temperature range. A rough estimate of the salt content in the inclusions was calculated based on the ice melting temperature by using a two-component water–salt system (NaCl–H$_2$O) through the equivalent of NaCl [29]. The predominant salt in the aqueous solution of inclusions was determined based on the eutectic temperature, which characterizes the water–salt system [30]. The gas composition of individual FIs was determined using Raman spectroscopy at the Institute of Geology and Mineralogy of the Siberian Branch of the Russian Academy of Sciences (Novosibirsk, Russia). A single-channel LabRam HR 800 Raman spectrometer with a Horiba Scientific Symphony II semiconductor detector and an Olympus BX-41 confocal microscope was used. The CVI MellesGirot laser with a wavelength of 514 nm and output power of 50–30 mW was used as excitation source. Water extracts from quartz grains containing FIs were analyzed on an atomic emission spectrometer, the Agilent 4100 MP-AES, by N.O. Vdovenko in NEISRI FEB RAS (Magadan, Russia).

## 4. Results

### 4.1. Deposit Geology

Inkur Deposit

The deposit frames the Pervomaisky granite-porphyry massif in the form of a semicircle on the southwestern, western, and northwestern sides (see Figure 1, black dotted line). The ore veinlets forming the stockwork lie in quartz diorites of the Modonkul massif and partially in metamorphosed sedimentary-volcanogenic rocks of the Khokhyurta suite. The stockwork extends for approximately 2500 m with a width of 800–850 m and is explored to a depth of 470–500 m.

Detailed structural studies of the Inkur stockwork revealed a radial-concentric structure of the deposit associated with the formation of a large arched uplift in quartz diorites under the influence of vertical stresses—in particular, the pressures of magmatic masses from the deep parts of the alleged focus. Under these conditions, radial and concentric cracks, where ore and vein minerals deposited, opened simultaneously [20].

The ore stockwork of the Inkur deposit (Figure 2a, Inkur deposit quarry) is composed of numerous quartz and quartz-muscovite veinlets with ore mineralization (Figure 2b). All veinlets are characterized by variable thickness. Ore mineralization is distributed in the stockwork very unevenly. Most hubnerite-rich sections (see Figure 2c) are marked in the southern and northern flanks, and the central part is characterized by relatively weak mineralization. In addition, areas with high content of wolframite spatially tend to veinlet swells [13]. Most of the W-bearing veinlets have a thickness of 1–3 cm (see Figure 2c), and their length, as a rule, reaches 1–5 m. The boundaries of the veinlets with host rocks are quite sharp. The presence of minerals (muscovite, hubnerite) growing from the walls of the veinlets to the central parts (see Figure 2c), as well as the sharpness of the boundaries of the veinlets with the host rocks, allowed their determination as crack-seal veinlets.

The host rock in the exocontacts of the ore veinlets intensively altered into quartz-sericite rocks (beresites); moreover, both granitoids and metasandstones of the volcanogenic-sedimentary sequence, as well as dike formations, are susceptible to quartz-sericite alterations. Metasomatic zones form vein-like and lenticular bodies with indistinct borders; in terms of thickness, they are from 1–5 cm to 10–50 m in size [16]. Secondary alterations consist of a quartz-muscovite assemblage with pyrite, carbonate, and fluorite, replacing the primary rocks, and with the distance from the veinlets, the intensity of metasomatism decreases.

### 4.2. Kholtoson Deposit

The Kholtoson deposit is located west of the above-described Inkur stockwork at a considerable distance (about 1 km) from the Pervomaisky granite massif (see Figure 1) within the contact part of the Modonkul intrusive of quartz diorites. The Modonkul intrusive is dissected by granite-syenogranite Upper Paleozoic plutons of the Daban series. One of the massifs is exposed in the southern part of the ore field, gently (at an angle of 15–20°) sinking to the north under the Kholtoson deposit [15].

The deposit was explored to a depth of 700–1000 m. In total, there are more than 200 known quartz-sulfide-hubnerite veins (Figure 2d, red dotted line). The veins lie mostly in diorites of the Modonkul massif and have a gentle drop to the south and the southwest.

Ore veins have a wide range of thicknesses, from tenths to 3–4 m, in swells—up to 15 m at average thickness of ~0.7–0.9 m (Figure 2d). About 80 veins are exhausted and characterized by economic parameters of W mineralization [20,21]. The average content of $WO_3$ is ~0.77 wt%. Now upper horizons of the main industrial veins are exhausted. Ore bodies of the Kholtoson deposit are represented by quartz-hubnerite vein sections, enriched with other ore minerals (sulfides and sulfosalts).

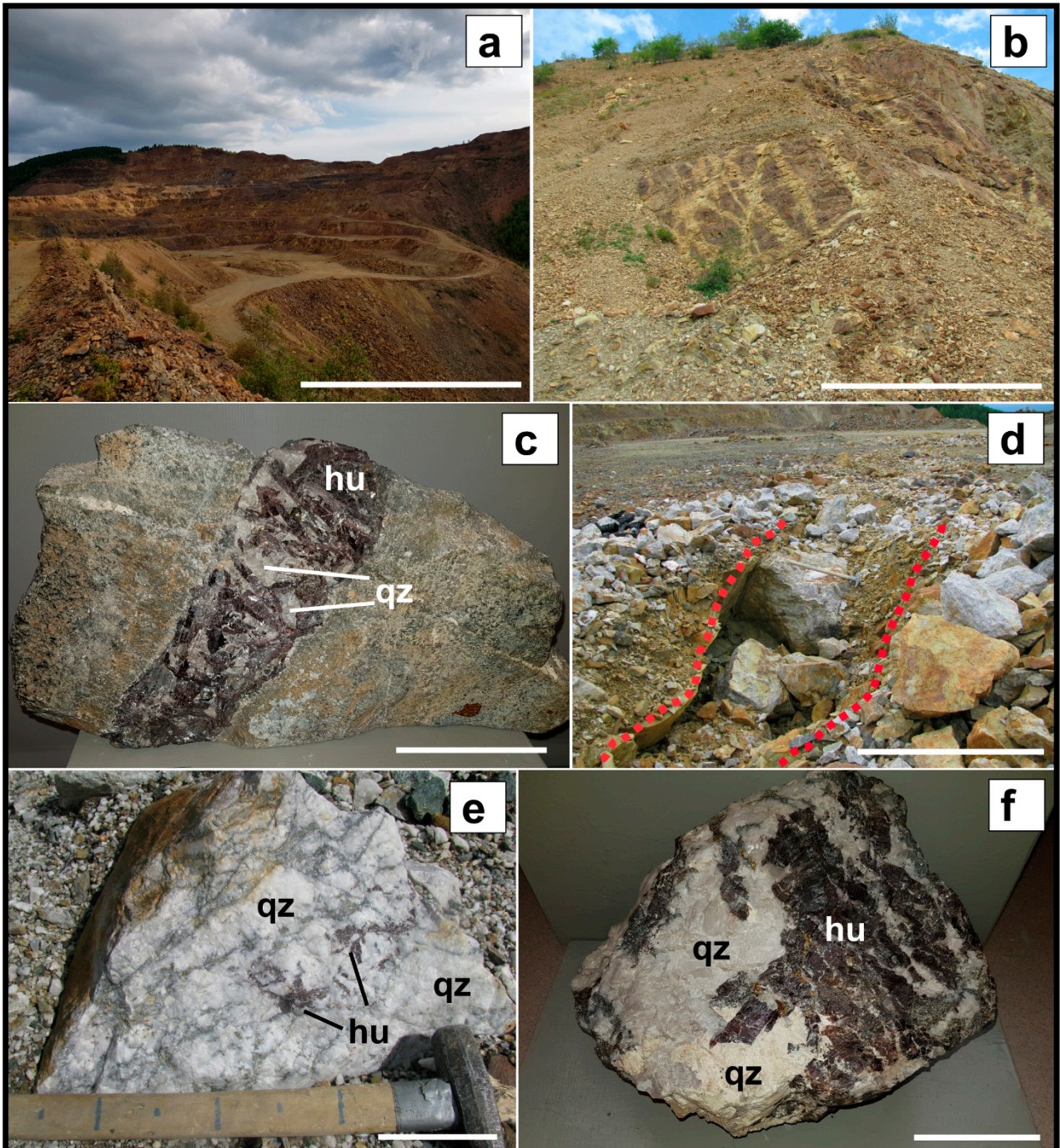

**Figure 2.** (**a**) General view of the Inkur quarry; (**b**) Inkur stockwork; (**c**) a hubnerite-rich quartz veinlet; (**d**) quartz-hubnerite veins of the Kholtoson deposit (red dotted line); (**e**) a sample of a quartz-sulfide vein with hubnerite (Kholtoson deposit); (**f**) rich quartz-hubnerite vein of the Kholtoson deposit. The samples in figures (**c**,**f**) are from the Geological Museum of the Buryat Scientific Center SB RAS, Ulan-Ude. Abbreviations of the minerals: hu—hubnerite, qz—quartz. Scale bar length: (**a**,**b**) 5 m, (**c**,**f**) 5 cm, (**d**) 1 m, (**e**) 10 cm.

## 5. Ore Mineralogy

### 5.1. Inkur Deposit

Gangue minerals composing ore veinlets of the Inkur deposit are represented by quartz, muscovite, fluorite, K-feldspar, and rare beryl (Table 1). The main gangue mineral, quartz (~50–90%), is a dominant mineral that has formed at all stages of mineral formation from early to late. Muscovite (~5–30%) has a white-greenish color, is mainly located along

the selvage parts of the veinlets (see Figure 2c), and tends to grow perpendicularly from the walls to the central parts; rarer scales of muscovite in the central parts of the veinlets are also observed. Veinlet selvages of low-thickness ($\leq$1 cm) are composed of relatively coarse-grained muscovite. A special feature of the muscovite chemical composition is its high F content (1.74–3.16% wt.%, Table 2).

**Table 1.** Bulk mineral composition of quartz-hubnerite veinlets of the Inkur and quartz-sulfide-hubnerite veins of the Kholtoson deposits.

| Assemblage | | Inkur Deposit | Kholtoson Deposit |
|---|---|---|---|
| | | **Minerals** | |
| Gangue minerals | Major | Quartz $SiO_2$ | Quartz $SiO_2$ |
| | Minor | Fluorite $CaF_2$ | Fluorite $CaF_2$ |
| | | K-feldspar $KAlSi_3O_8$ | K-feldspar $KAlSi_3O_8$ |
| | | Muscovite $KAl_2[AlSi_3O_{10}](OH)_2$ | Muscovite $KAl_2[AlSi_3O_{10}](OH)_2$ |
| Ore minerals | Major | Hubnerite $MnWO_4$ | Hubnerite $MnWO_4$ |
| | | Pyrite $FeS_2$ | Pyrite $FeS_2$ |
| | | Chalcopyrite $CuFeS_2$ | Chalcopyrite $CuFeS_2$ |
| | Minor | Sphalerite $ZnS$ | Sphalerite $ZnS$ |
| | | Galena $PbS$ | Galena $PbS$ |
| | | Tetrahedrite $Cu_3SbS_3$ | Tetrahedrite $Cu_3SbS_3$ |
| | | Aikinite $PbCuBiS_3$ | Aikinite $PbCuBiS_3$ |
| | | Molybdenite $MoS_2$ | Scheelite—$CaWO_4$ |
| | | Scheelite $CaWO_4$ | Hessite $Ag_2Te$ |
| | - | Cassiterite $SnO_2$ | Stannite $Cu_2FeSnS_4$ |
| | - | - | Siderite $FeCO_3$ Rhodochrosite—$MnCO_3$ |
| | Rare | Hessite $Ag_2Te$ | Schapbachite $Ag_{0.4}Bi_{0.4}S$ |
| | | Bornite $Cu_5FeS_4$ | Bornite $Cu_5FeS_4$ |
| | | Beryl $Al_2[Be_3(Si_6O_{18})]$ | Matildite—$AgBiS_2$ |
| - | | - | Unknown phases $Cu_2PbS_3$, $Cu_2Pb_3S_5$ |
| Secondary minerals | | Anglesite $PbSO_4$ | Anglesite $PbSO_4$ |
| | | Rosnbergite(?)$AlF[F_{0.5}(H_2O)_{0.5}]_4 \cdot H_2O$ | Cerussite $PbCO_3$ |

**Table 2.** Chemical composition of muscovite from Inkur and Kholtoson deposits.

| № | $SiO_2$ | $TiO_2$ | $Al_2O_3$ | FeO | MnO | MgO | $Na_2O$ | $K_2O$ | F | $H_2O$ | Total |
|---|---|---|---|---|---|---|---|---|---|---|---|
| | | | | | **Inkur Deposit** | | | | | | |
| 1. | 47.22 | - | 28.57 | 0.42 | 0.61 | 2.44 | - | 12.08 | 3.06 | 4.5 | 99.11 |
| 2. | 47.48 | 0.33 | 28.52 | 0.46 | 0.53 | 2.72 | - | 12.36 | 2.43 | 4.5 | 99.42 |
| 3. | 47.99 | - | 28.84 | 0.33 | 0.68 | 2.27 | - | 12.30 | 2.35 | 4.5 | 99.48 |
| 4. | 48.74 | - | 26.90 | - | 0.50 | 3.55 | - | 12.55 | 3.16 | 4.5 | 100.13 |
| 5. | 48.50 | - | 29.44 | - | 0.64 | 2.46 | - | 12.01 | 3.00 | 4.5 | 100.78 |
| 6. | 49.05 | - | 27.35 | - | 0.94 | 2.93 | - | 12.23 | 3.13 | 4.5 | 100.22 |
| 7. | 47.78 | - | 28.14 | 1.31 | 1.02 | 2.34 | - | 12.33 | 2.14 | 4.5 | 99.92 |
| 8. | 48.27 | - | 29.00 | - | 0.73 | 2.56 | - | 12.27 | 2.43 | 4.5 | 100.12 |

**Table 2.** *Cont.*

| № | SiO$_2$ | TiO$_2$ | Al$_2$O$_3$ | FeO | MnO | MgO | Na$_2$O | K$_2$O | F | H$_2$O | Total |
|---|---|---|---|---|---|---|---|---|---|---|---|
| | | | | | **Inkur Deposit** | | | | | | |
| 9. | 47.87 | - | 28.19 | 0.96 | 0.37 | 2.38 | - | 12.67 | 2.80 | 4.5 | 99.97 |
| 10. | 47.30 | - | 31.69 | 0.93 | 0.51 | 1.48 | 0.32 | 12.29 | 1.74 | 4.5 | 100.99 |
| 11. | 47.96 | - | 30.20 | - | 0.46 | 2.15 | 0.32 | 12.31 | 2.60 | 4.5 | 100.59 |
| | | | | | **Kholtoson Deposit** | | | | | | |
| 1. | 47.96 | - | 29.93 | 0.44 | 0.72 | 2.26 | - | 11.56 | 1.96 | 4.5 | 99.33 |
| 2. | 46.38 | - | 29.97 | 0.68 | 0.59 | 1.89 | 0.38 | 11.84 | 2.48 | 4.5 | 98.71 |
| 3. | 48.63 | 0.44 | 29.91 | 0.00 | 0.54 | 2.64 | 0.32 | 11.68 | 1.74 | 4.5 | 100.08 |
| 4. | 48.14 | 0.35 | 29.46 | 0.48 | 0.4 | 2.5 | - | 11.58 | 2.69 | 4.5 | 100.1 |
| 5. | 48.18 | - | 27.4 | 1.18 | 0.92 | 2.39 | 0.39 | 11.3 | 1.73 | 4.5 | 97.99 |
| 6. | 48.8 | - | 29.93 | 0.44 | 1.38 | 2.16 | - | 11.58 | 2.47 | 4.5 | 101.26 |
| 7. | 46.81 | - | 31.5 | 0.37 | 0.67 | 1.66 | - | 12.02 | 2.52 | 4.5 | 100.05 |
| 8. | 47.41 | - | 29.19 | 0 | 1.15 | 2.17 | - | 12.00 | 3.21 | 4.5 | 99.63 |
| 9. | 50.83 | - | 25.74 | 0.00 | 0.63 | 3.73 | - | 11.22 | 2.50 | 4.5 | 98.84 |

Notes: H$_2$O content is calculated according to mineral stoichiometry.

**Table 3.** Chemical composition of W-bearing minerals.

| № | Fe | Mn | Ca | W | O | Total |
|---|---|---|---|---|---|---|
| | | | **Inkur Deposit** | | | |
| 1. | - | 18.69 | - | 63.44 | 16.94 | 99.08 |
| 2. | - | 18.06 | - | 63.84 | 17.61 | 99.50 |
| 3. | 0.61 | 17.76 | - | 62.82 | 18.05 | 99.25 |
| 4. | 0.59 | 17.78 | - | 63.13 | 18.40 | 99.88 |
| 5. | - | 18.78 | - | 63.03 | 17.84 | 99.64 |
| 6. | - | 18.31 | - | 63.97 | 17.43 | 99.72 |
| | | | **Kholtoson Deposit** | | | |
| 7. | | 18.66 | - | 63.01 | 18.56 | 100.23 |
| 8. | 0.49 | 19.34 | - | 62.48 | 17.83 | 100.14 |
| 9. | 0.53 | 18.24 | - | 63.37 | 17.64 | 99.78 |
| 10. | 0.46 | 18.70 | - | 62.97 | 17.73 | 99.86 |
| 11. | 0.98 | 17.86 | - | 62.86 | 18.33 | 100.03 |
| 12. | 0.87 | 18.15 | - | 62.96 | 17.67 | 99.64 |
| 13. | - | 18.41 | - | 64.21 | 16.53 | 99.16 |
| 14. | 0.64 | 18.42 | - | 62.73 | 18.04 | 99.82 |
| 15. | - | 18.52 | - | 64.16 | 17.10 | 99.78 |
| 16. | - | - | 14.18 | 66.00 | 20.54 | 100.72 |
| 17. | - | - | 14.11 | 67.30 | 19.20 | 100.61 |
| 18. | - | - | 14.24 | 66.99 | 18.83 | 100.06 |
| 19. | - | - | 14.37 | 66.69 | 18.56 | 99.62 |
| 20. | 0.93 | - | 13.68 | 65.09 | 19.34 | 99.04 |
| 21. | 1.25 | - | 14.15 | 62.81 | 22.27 | 100.47 |

Notes: 1–15—hubnerite, 16–21—scheelite; empty cell—below detection limit.

In the veinlets, distribution of ore mineralization is extremely uneven, from visually ore-free to ore-mineral-rich sections (up to ~40–50%).

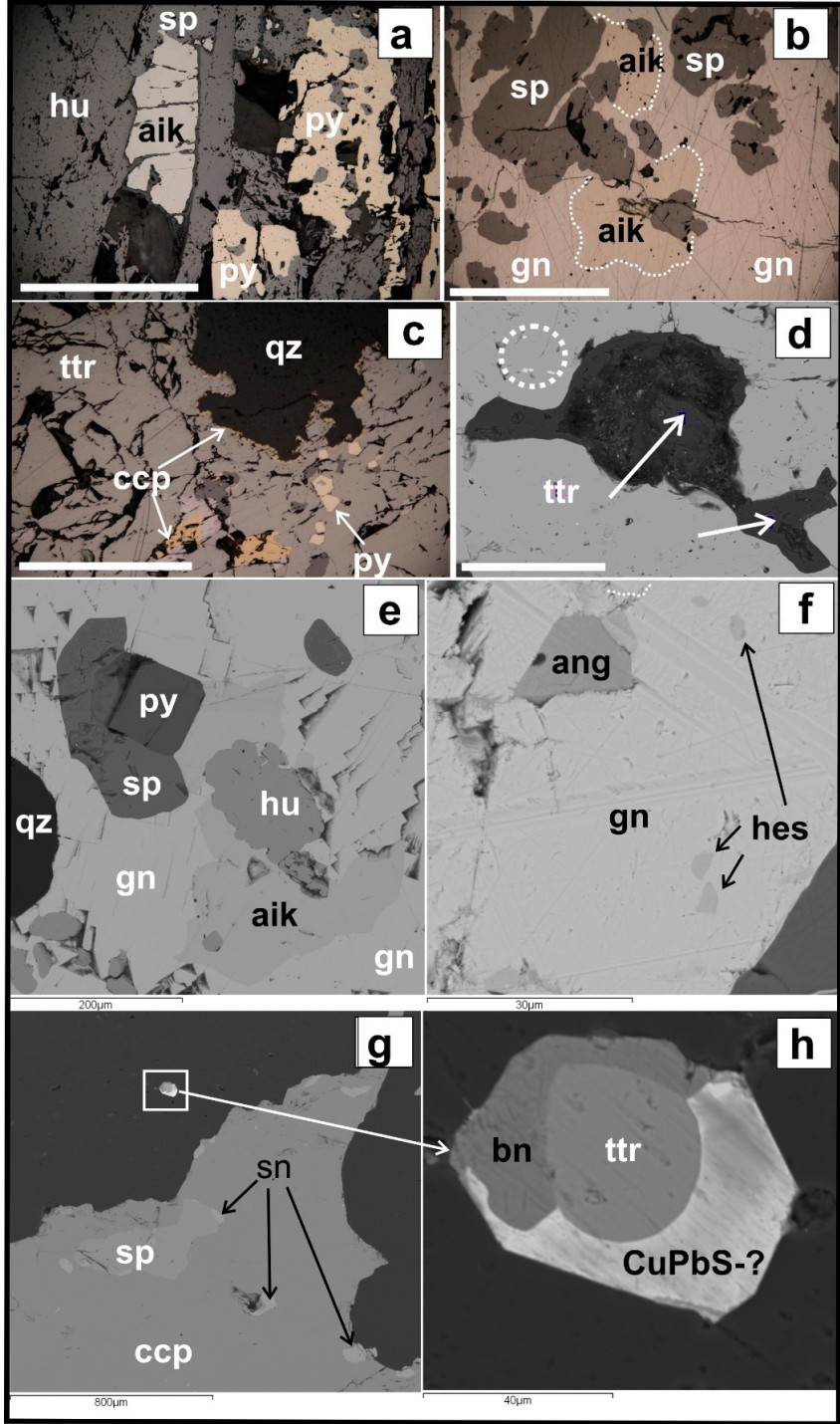

**Figure 3.** Petrographic characteristics of the ore minerals. Photomicrographs of polished sections
(**a**–**c**) and BSE images (**d**–**h**) of the Inkur (**a**–**d**) and the Kholtoson (**e**–**h**) deposits. (**a**) interrelationships
of hubnerite, aikinite, and sphalerite, corroded pyrite grains; (**b**) sphalerite and aikinite (white dotted
line) grains in the galena; (**c**) chalcopyrite and pyrite in tetrahedrite aggregate; (**d**) rosenbergite (white
arrows) enclosed in tetrahedrite, with microinclusions (dotted line) of hessite; (**f**) inclusions of hessite
(arrows) in galena and anglesite segregation; (**g**) sphalerite and stannite enclosed in chalcopyrite
aggregate; (**h**) intergrowths of bornite, tetrahedrite, and unknown CuPbS phase. Scale bar length:
(**a**–**c**) 1 cm, (**d**) 200 μm. Abbreviations of the minerals: hu—hubnerite, sp—sphalerite, py—pyrite, gn—
galena, ccp—chalcopyrite, aik—aikinite, ttr—tetrahedrite, qz—quartz, ang—anglesite, hes—hessite,
sn—stannite, bn—bornite, rs—rosenbergite.

The main ore mineral (wolframite) is represented by a manganese variety, hubnerite, with Mn content of 17.76–18.78 wt.% and Fe of 0.59–0.61 wt.% in single grains, as determined by EDS analysis (Table 3). Hubnerite tends to forms elongated pole-shaped or columnar crystals (see Figures 2c and 3a) of reddish-brown color, growing in sections from the walls of the veinlets, or clusters of irregular-shaped crystals. Less often, hubnerite occurs in the form of individual grains of irregular shape.

At the deposit, there are widely developed sulfide minerals, among which pyrite, sphalerite, galena and chalcopyrite predominate (Figure 3).

Pyrite is present in two generations, differing in morphology. Pyrite of the first generation (Figure 3a) occurs as intensely corroded grains, replaced by later minerals—galena, sphalerite, tetrahedrite, etc. Pyrite of the second generation is present as cubic crystals associated with tetrahedrite.

Chalcopyrite is found intergrown with galena and sphalerite, as well as rims on the edges of tetrahedrite grains (see Figure 3c).

Galena is often found associated with sphalerite and aikinite (Figure 3b). Galena is characterized by the presence of Ag and Bi impurities, the contents of which reach concentrations as high as 7.24 wt.% Ag and 6.69 wt.% Bi; hence, it can be referred to as Ag-Bi-containing galena (Table 4).

**Table 4.** Chemical composition of ore minerals from the Inkur deposit.

| № | Fe | Co | Cu | Zn | Ag | Cd | Sn | Mo | Pb | Bi | Sb | Te | As | S | O | Total |
|---|----|----|----|----|----|----|----|----|----|----|----|----|----|----|----|-------|
| 1. | - | - | 11.09 | - | - | - | - | - | 40.94 | 32.46 | - | - | - | 15.77 | - | 100.27 |
| 2. | - | - | 10.71 | - | - | - | - | - | 41.51 | 31.83 | - | - | - | 15.52 | - | 99.58 |
| 3. | - | - | 10.60 | - | - | - | - | - | 42.72 | 30.84 | - | - | - | 15.01 | - | 99.16 |
| 4. | - | - | 10.63 | - | - | - | - | - | 41.84 | 31.36 | - | - | - | 15.59 | - | 99.41 |
| 5. | - | - | - | - | - | - | 71.88 | - | - | - | - | - | - | - | 27.54 | 99.41 |
| 6. | 30.19 | - | 34.81 | - | - | - | - | - | - | - | - | - | - | 34.99 | - | 99.99 |
| 7. | 30.01 | - | 35.05 | - | - | - | - | - | - | - | - | - | - | 34.06 | - | 99.13 |
| 8. | 30.19 | - | 34.63 | - | - | - | - | - | - | - | - | - | - | 34.13 | - | 98.95 |
| 9. | - | - | - | - | 3.77 | - | - | - | 77.72 | 5.31 | - | - | - | 12.52 | - | 99.31 |
| 10. | - | - | - | - | 6.69 | - | - | - | 76.35 | 3.32 | - | - | - | 12.33 | - | 98.69 |
| 11. | - | - | - | - | 3.5 | - | - | - | 77.97 | 6.69 | - | - | - | 12.28 | - | 100.44 |
| 12. | - | - | - | - | 2.61 | - | - | - | 80.83 | 4.15 | - | - | - | 12.76 | - | 100.36 |
| 13. | - | - | - | - | 2.58 | - | - | - | 81.01 | 4.01 | - | - | - | 11.92 | - | 99.52 |
| 14. | - | - | - | - | 3.66 | - | - | - | 78.52 | 5.23 | - | - | - | 12.36 | - | 99.77 |
| 15. | - | - | - | - | 2.88 | - | - | - | 80.25 | 5.4 | - | - | - | 12.49 | - | 101.02 |
| 16. | - | - | - | - | 7.24 | - | - | - | 76.75 | 4.09 | - | - | - | 12.14 | - | 100.22 |
| 17. | - | - | - | - | - | - | - | - | 87.23 | - | - | - | - | 12.26 | - | 99.49 |
| 18. | - | - | - | - | - | - | - | - | 86.99 | - | - | - | - | 12.23 | - | 99.22 |
| 19. | - | - | - | - | 63.69 | - | - | - | - | - | - | 36.85 | - | - | - | 100.54 |
| 20. | - | - | - | - | 61.35 | - | - | - | - | - | - | 37.02 | - | - | - | 98.37 |
| 21. | - | - | - | - | 58.87 | - | - | - | - | - | - | 35.23 | - | - | - | 94.1 |
| 22. | - | - | - | - | 58.68 | - | - | - | - | - | - | 39.72 | - | - | - | 98.4 |
| 23. | - | - | - | - | 57.81 | - | - | - | - | - | - | 40.29 | - | - | - | 98.1 |
| 24. | - | - | - | - | - | - | 55.82 | - | - | - | - | - | - | 43.95 | - | 99.76 |

**Table 4.** *Cont.*

| № | Fe | Co | Cu | Zn | Ag | Cd | Sn | Mo | Pb | Bi | Sb | Te | As | S | O | Total |
|---|----|----|----|----|----|----|----|----|----|----|----|----|----|---|---|-------|
| 25. | 46.64 | - | - | - | - | - | - | - | - | - | - | - | - | 52.51 | - | 99.14 |
| 26. | 46.88 | - | - | - | - | - | - | - | - | - | - | - | - | 52.7 | - | 99.58 |
| 27. | 47.5 | - | - | - | - | - | - | - | - | - | - | - | - | 53.44 | - | 100.94 |
| 28. | 46.6 | - | - | - | - | - | - | - | - | - | - | - | - | 52.78 | - | 99.38 |
| 29. | 46.49 | 0.48 | - | - | - | - | - | - | - | - | - | - | - | 52.59 | - | 99.56 |
| 30. | 46.34 | 0.56 | - | - | - | - | - | - | - | - | - | - | - | 52.24 | - | 99.14 |
| 31. | - | - | - | 66.66 | - | 0.58 | - | - | - | - | - | - | - | 33.06 | - | 100.3 |
| 32. | - | - | - | 67.42 | - | 0.66 | - | - | - | - | - | - | - | 32.78 | - | 100.85 |
| 33. | - | - | - | 65.8 | - | 0.53 | - | - | - | - | - | - | - | 32.81 | - | 99.14 |
| 34. | - | - | - | 66.45 | - | 0.69 | - | - | - | - | - | - | - | 32.73 | - | 99.86 |
| 35. | - | - | - | 66.15 | - | 0.62 | - | - | - | - | - | - | - | 32.2 | - | 98.97 |
| 36. | 1.21 | - | - | 64.29 | - | 0.67 | - | - | - | - | - | - | - | 32.57 | - | 98.74 |
| 37. | - | - | - | 65.57 | - | 0.96 | - | - | - | - | - | - | - | 32.59 | - | 99.12 |
| 38. | - | - | - | 64.97 | - | 0.94 | - | - | - | - | - | - | - | 32.45 | - | 98.36 |
| 39. | 1.48 | - | 38.21 | 6.53 | 0.85 | - | - | - | - | 1.59 | 18.23 | - | 6.63 | 24.85 | - | 98.37 |
| 40. | 0.45 | - | 39.98 | 7.26 | - | - | - | - | - | - | 22.14 | - | 4.50 | 25.93 | - | 100.26 |
| 41. | - | - | 39.07 | 7.26 | 0.85 | - | - | - | - | 1.52 | 21.83 | - | 5.27 | 25.84 | - | 101.64 |
| 42. | - | - | 38.79 | 6.83 | 0.95 | - | - | - | - | - | 22.07 | - | 4.95 | 25.67 | - | 99.25 |
| 43. | - | - | 39.37 | 7.44 | 0.8 | - | - | - | - | - | 21.04 | - | 5.04 | 25.73 | - | 99.43 |
| 44. | 0.52 | - | 39.36 | 7.84 | 0.84 | - | - | - | - | - | 22.20 | - | 5.08 | 25.35 | - | 101.2 |
| 45. | 0.52 | - | 38.45 | 7.62 | - | - | - | - | - | - | 19.76 | - | 6.57 | 25.96 | - | 98.87 |
| 46. | - | - | 38.07 | 7.58 | 1.08 | - | - | - | - | - | 21.93 | - | 4.80 | 25.31 | - | 98.77 |
| 47. | 0.38 | - | 38.84 | 9.09 | - | - | - | - | - | - | 18.59 | - | 7.07 | 25.53 | - | 99.51 |
| 48. | 0.46 | - | 39.22 | 7.07 | 0.65 | - | - | - | - | - | 17.21 | - | 8.80 | 25.71 | - | 99.12 |
| 49. | - | - | 37.89 | 7.09 | 1.21 | - | - | - | - | - | 22.07 | - | 4.95 | 25.48 | - | 98.69 |

Notes: 1–4—aikinite; 5—cassiterite; 6–8—chalcopyrite; 9–18—galena; 19–23—hessite; 24—molybdenite; 25–30—pyrite; 31–38—sphalerite; 39–49—tetrahedrite.

Sphalerite (Figure 3a,b) does not contain Fe; Cd (0.49–0.96 wt.%) was determined to be the only impurity in it. In sphalerite, there are noted single inclusions of small cassiterite crystals.

Bornite is present as rare single grains in the ores.

Molybdenite was observed as single small lamellas in the marginal parts of the veinlets in the aggregates of muscovite and quartz. In the veinlets, at deeper horizons, previous studies also noted molybdenite, forming a fine impregnation in microcline and muscovite at the veinlet edges. It is believed that this is the redeposited molybdenite from earlier molybdenite-containing quartz-albite veinlets [14,15].

Along with molybdenite, scheelite was also observed both at early and later stages during the replacement of hubnerite crystals.

Aikinite and tetrahedrite predominate among the sulfosalts.

Aikinite composes mainly irregular-shaped grains with barely discernible boundaries in the galena grains (see Figure 3b, white dotted line).

Tetrahedrite forms relatively large aggregates of irregular shape (see Figure 3c,d) or single grains of isometric shape. SEM EDS analysis showed Sb content of 17.21–22.20 wt.% and As content of 4.50–8.80 wt.%. Tetrahedrite also contains Fe (0.38–1.48 wt.%), Zn (6.53–9.09 wt.%), Ag (0.65–1.21 wt.%), and in some grains, Bi impurities (up to

1.59 wt.%) were found (see Table 4). Silver telluride—hessite (see Figure 3d, white dotted line) was present as numerous microinclusions in tetrahedrite.

In one tetrahedrite grain, a rare unusual mineral from the halide class—aluminum hydrofluoride was identified; its composition is similar to rosenbergite (AlF[F$_{0.5}$(H$_2$O)$_{0.5}$]$_4$·H$_2$O) (see Figure 3d), and it forms irregular grains with rounded outlines.

Sulfosalts form paragenetic assemblages with sulfides—galena, sphalerite, chalcopyrite, and pyrite, which allows us to conclude that the formation of these minerals was synchronous.

Supergene minerals include anglesite and covelline (Figure 4), filling thin late cracks or voids.

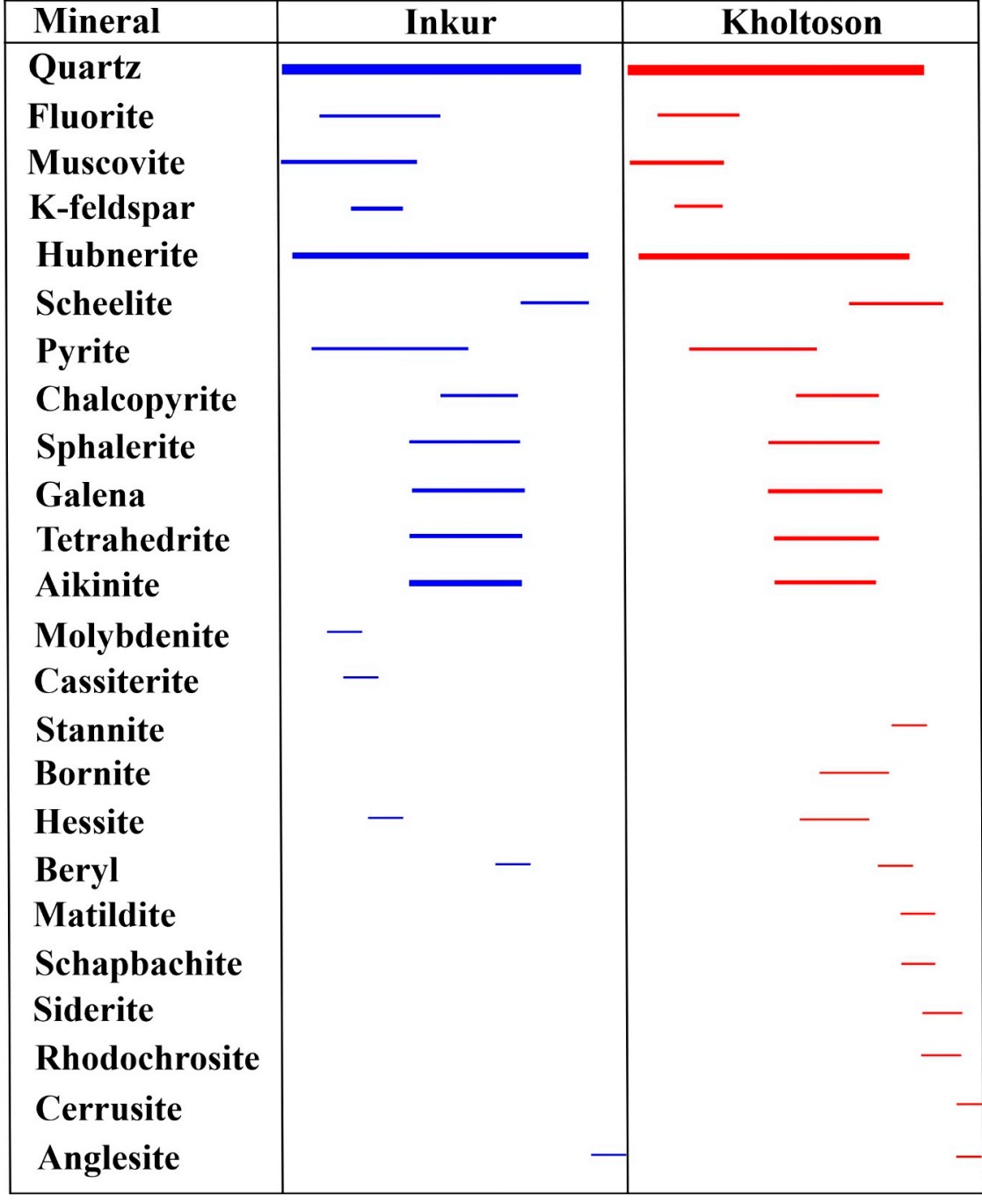

**Figure 4.** General paragenetic sequence of the Inkur and the Kholtoson deposits.

In addition, such minerals as topaz ($Al_2[SiO_4](F, OH)_2$), lindströmite ($Pb_3Cu_3Bi_7S_{15}$), hammarite ($Pb_2Cu_2Bi_4S_9$), sulfobismuthite of Cu and of Ag, and rare silver and gold tellurides (petzite and sylvanite) in assemblage with native gold were identified by previous researchers [31].

### 5.2. Kholtoson Deposit

The Kholtoson deposit is represented by quartz-hubnerite veins with sulfide mineralization. The mineral composition of the Kholtoson deposit is identical to the mineral composition of the Inkur deposit (Table 1). Quartz is the main gangue mineral, and among the ore minerals, there are hubnerite, pyrite, chalcopyrite, galena, sphalerite, scheelite, etc. The gangue minerals are represented by mainly milky-white quartz, and to a lesser extent, K-feldspar, micas, and rare fluorite and beryl. In thin sections, quartz grains are characterized by wavy or mosaic extinction. In some areas, there is an intense sericitization of feldspars, as well as thin sericite veinlets in quartz grains. Wall-rock alterations are represented by greisenization, where quartz-muscovite aggregates occur. Muscovite, as in the veinlets of the Inkur deposit, is characterized by increased content of F (1.73–3.21 wt.%) (see Table 2).

In some areas, in ore quartz veins, there are carbonates, represented by rhodochrosite and siderite, which were found mainly as xenomorphic aggregates of irregular shape; they are often connected to quartz grain contacts, or present in the form of thin late veinlets filling small cracks.

Quartz is a dominant mineral that is formed at all stages of mineral formation from early to late (Figure 4). Aggregates of K-feldspar are in close association with quartz, which indicates their near-simultaneous formation.

Quartz-hubnerite veins are mainly composed of grains of light, light-gray, or milky-white quartz with single hubnerite crystals (Figure 2e) or enriched areas with larger amounts of hubnerite aggregates (Figure 2f). Such areas contain coarse-grained aggregates of hubnerite in assemblage with other ore minerals (pyrite, galena, sphalerite, tetrahedrite) (see Table 1).

The main ore mineral of the Kholtoson deposit is hubnerite, with Mn content of 17.86–19.34 wt.% and Fe content of 0.49–0.98 wt.% (see Table 3). Hubnerite is represented by elongated euhedral crystals, less often by irregularly shaped aggregates with characteristic bright red, brownish, or brown colors (Figure 2f). Distinct idiomorphism of hubnerite crystals and aggregates of single grains indicate its deposition at both the early and later stages (Figure 4).

In addition to hubnerite, scheelite is also present in smaller amounts. It composes aggregates forming rims on hubnerite grains. Therefore, scheelite is a late mineral replacing hubnerite.

Pyrite often forms euhedral crystals that are close to cubic in shape (Figure 3e). Pyrite grains are often corroded by other ore minerals, more often by galena which indicates earlier formation of pyrite. In some samples, accumulations of pyrite grains were observed.

Galena mainly forms aggregates of irregular shape more often in association with sphalerite, less often with chalcopyrite and tetrahedrite (Figure 3e). In some areas, it fills cracks in hubnerite crystals. Galena contains impurities of Ag (0.83–1.99 wt.%) and Bi (1.95–3.15 wt.%) (Table 5).

**Table 5.** Chemical composition of ore minerals from the Kholtoson deposit.

| № | Fe | Co | Cu | Zn | Ag | Cd | Sn | Pb | Bi | Sb | Te | As | S | Total |
|---|----|----|----|----|----|----|----|----|----|----|----|----|----|----|
| 1. | - | - | 11.19 | - | - | - | - | 41.67 | 31.90 | - | - | - | 15.15 | 99.91 |
| 2. | - | - | 11.60 | - | - | - | - | 42.87 | 30.86 | - | - | - | 14.92 | 100.25 |
| 3. | - | - | 11.67 | - | - | - | - | 42.00 | 31.29 | - | - | - | 15.15 | 100.13 |
| 4. | - | - | 10.82 | - | - | - | - | 42.24 | 31.64 | - | - | - | 15.26 | 99.96 |
| 5. | - | - | 30.82 | - | - | - | - | 48.52 | - | - | - | - | 23.12 | 102.46 |
| 6. | - | - | 30.69 | - | - | - | - | 47.94 | - | - | - | - | 22.75 | 101.38 |
| 7. | - | - | 13.89 | - | - | - | - | 69.38 | - | - | - | - | 16.49 | 99.75 |
| 8. | - | - | 13.47 | - | - | - | - | 69.23 | - | - | - | - | 16.32 | 99.01 |
| 9. | 11.94 | - | 59.43 | - | - | - | - | - | - | - | - | - | 28.03 | 99.39 |
| 10. | 11.43 | - | 59.09 | - | - | - | - | - | - | - | - | - | 28.71 | 99.24 |
| 11. | 30.81 | - | 34.86 | - | - | - | - | - | - | - | - | - | 34.99 | 100.65 |
| 12. | 30.21 | - | 33.83 | - | - | - | - | - | - | - | - | - | 35.34 | 99.38 |
| 13. | 30.57 | - | 33.51 | - | - | - | - | - | - | - | - | - | 35.21 | 99.30 |
| 14. | - | - | - | - | 1.44 | - | - | 82.62 | 3.15 | - | - | - | 12.32 | 99.54 |
| 15. | - | - | - | - | 0.83 | - | - | 85.46 | 1.95 | - | - | - | 11.93 | 100.16 |
| 16. | - | - | - | - | - | - | - | 87.7 | - | - | - | - | 12.81 | 100.51 |
| 17. | - | - | - | - | 1.34 | - | - | 83.85 | 3.08 | - | - | - | 12.70 | 100.97 |
| 18. | - | - | - | - | 1.99 | - | - | 83.61 | 2.88 | - | - | - | 12.49 | 100.97 |
| 19. | - | - | - | - | 1.26 | - | - | 84.09 | 2.29 | - | - | - | 12.63 | 100.27 |
| 20. | - | - | - | - | 63.79 | - | - | - | - | - | 37.19 | - | - | 100.98 |
| 21. | - | - | - | - | 63.18 | - | - | - | - | - | 37.19 | - | - | 100.37 |
| 22. | 46.63 | 0.54 | - | - | - | - | - | - | - | - | - | - | 52.34 | 99.51 |
| 23. | 47.46 | - | - | - | - | - | - | - | - | - | - | - | 53.08 | 100.54 |
| 24. | 47.11 | - | - | - | - | - | - | - | - | - | - | - | 53.01 | 100.13 |
| 25. | 46.52 | 0.52 | - | - | - | - | - | - | - | - | - | - | 52.85 | 99.89 |
| 26. | - | - | - | - | 27.43 | - | - | 10.05 | 46.12 | - | - | - | 15.98 | 99.58 |
| 27. | - | - | - | - | 27.73 | - | - | 5.05 | 50.86 | - | - | - | 15.91 | 99.55 |
| 28. | - | - | - | - | 28.77 | - | - | 6.06 | 50.59 | - | - | - | 16.49 | 101.92 |
| 29. | - | - | - | 65.91 | - | 1.34 | - | - | - | - | - | - | 31.99 | 99.23 |
| 30. | - | - | - | 66.38 | - | 1.15 | - | - | - | - | - | - | 32.15 | 99.68 |
| 31. | 0.41 | - | - | 65.46 | - | 0.95 | - | - | - | - | - | - | 32.2 | 99.02 |
| 32. | - | - | - | 67.11 | - | 1.13 | - | - | - | - | - | - | 32.11 | 100.34 |
| 33. | 0.34 | - | - | 65.36 | - | 1.1 | - | - | - | - | - | - | 32.66 | 99.46 |
| 34. | - | - | - | 66.26 | - | 1.18 | - | - | - | - | - | - | 32.83 | 100.28 |
| 35. | 0.3 | - | - | 66.42 | - | 0.99 | - | - | - | - | - | - | 32.31 | 100.03 |
| 36. | 0.55 | - | - | 65.59 | - | 0.64 | - | - | - | - | - | - | 32.87 | 99.64 |
| 37. | - | - | - | 66.61 | - | 0.8 | - | - | - | - | - | - | 33.01 | 100.41 |
| 38. | 8.52 | - | 37.46 | 7.16 | - | - | 17.79 | - | - | - | - | - | 29.07 | 100.01 |

**Table 5.** *Cont.*

| № | Fe | Co | Cu | Zn | Ag | Cd | Sn | Pb | Bi | Sb | Te | As | S | Total |
|---|---|---|---|---|---|---|---|---|---|---|---|---|---|---|
| 39. | 8.95 | - | 37.57 | 5.13 | - | - | 19.12 | - | - | - | - | - | 29.6 | 100.37 |
| 40. | 9.71 | - | 38.19 | 4.79 | - | - | 18.41 | - | - | - | - | - | 29 | 100.09 |
| 41. | 9.76 | - | 37.63 | 4.53 | - | - | 18.49 | - | - | - | - | - | 29.39 | 99.8 |
| 42. | 8.74 | - | 38.01 | 6.46 | - | - | 17.83 | - | - | - | - | - | 28.76 | 99.81 |
| 43. | 9.28 | - | 37.65 | 5.21 | - | - | 18.36 | - | - | - | - | - | 28.93 | 99.43 |
| 44. | 10.42 | - | 37.1 | 5.14 | - | - | 18.83 | - | - | - | - | - | 29.34 | 100.83 |
| 45. | - | - | 40.11 | 7.63 | - | - | - | - | - | 20.19 | - | 6.13 | 25.79 | 99.85 |
| 46. | - | - | 38.78 | 7.03 | - | - | - | 0.00 | 2.26 | 20.03 | - | 5.49 | 25.81 | 99.41 |
| 47. | - | - | 38.83 | 7.47 | 0.49 | - | - | 2.18 | - | 21.09 | - | 4.77 | 25.66 | 100.50 |

Notes: 1–4—aikinite; 5–6—unknown phase $Cu_2PbS_3$; 7–8—unknown phase $Cu_2Pb_3S_5$; 9–10—bornite; 11–13—chalcopyrite; 14–19—galena; 20–21—hessite; 22–25—pyrite; 26—schapbachite; 27–28—matildite; 29–37—sphalerite; 38–44—stannite; 45–47—tetrahedrite.

Sphalerite aggregates of mostly irregular shape (Figure 3g) form close intergrowth with galena, which apparently indicates their near-simultaneous formation. Sphalerite contains Cd (0.80–1.34 wt.%), and in some grains, Fe (0.30–0.55 wt.%) (see Table 5).

Chalcopyrite is observed as irregular-shaped aggregates (Figure 3g), or it often forms rims around pyrite grains; this suggests that chalcopyrite was formed later than pyrite. In some areas, the mineral is observed as small inclusions or fines in sphalerite aggregates.

Tetrahedrite forms relatively euhedral crystals of cubic or irregular shape in intergrowth with both aggregates of chalcopyrite and galena. This indicates that they deposited in veins at nearly the same time, or that the tetrahedrite is somewhat earlier due to more distinct idiomorphism. Additionally, it is observed as teardrop-shaped microinclusions (Figure 3h). In the tetrahedrite, the As content varies in the range 4.77–6.13 wt.%, and Zn in the range 7.03–7.63 wt.%. In some grains, there is an Ag impurity (0.49 wt.%).

Aikinite is present as irregular-shaped grains in galena aggregates (Figure 3e).

Stannite is observed as small solid inclusions in the aggregates of chalcopyrite and sphalerite (Figure 3 g) and, in addition, as small fines in the pyrite grains. A special feature of the stannite chemical composition is the increased content of Zn (4.53–7.16 wt.%) and Cu (up to 38.19 wt.%) and relatively decreased content of Fe (8.52–10.42 wt.%) and Sn (17.83–19.12 wt.%) compared to the stoichiometric composition.

Bornite (Figure 3h) and hessite (Figure 3f) were found as single grains in quartz and microinclusions in galena.

In galena there are schapbachite—Ag-Pb sulfobismutite ($Ag_{0.4}Pb_{0.2}Bi_{0.4}S$)—and Pb-bearing matildite ($AgBiS_2$) as rare inclusions (see Table 5).

Moreover, in the ores, there were identified unknown phases with formulas similar to $Cu_2PbS_3$ and $Cu_2Pb_3S_5$ (see Table 5), forming a heterogeneous aggregate in the intergrowth with bornite and tetrahedrite (Figure 3h).

Secondary supergene minerals are represented by anglesite and cerussite, often coinciding with cracks and voids, which indicates their late formation.

The general paragenetic sequence of the Inkur and the Kholtoson deposits is shown in Figure 4.

## 6. Fluid Inclusion Study

The data on the formation conditions and the salt composition of ore-forming solutions of the Inkur and Kholtoson W mineralization were obtained as a result of the study of fluid inclusions (FIs) hosted in quartz, fluorite, and hubnerite, and in a single case, of muscovite hosted in quartz-hubnerite veinlets and quartz-sulfide-hubnerite veins. In these minerals, we found inclusions with the most suitable sizes for studying by microthermometry methods.

Fluid inclusion analysis in wolframite and quartz was focused on fluid inclusion assemblages if the inclusions were trapped along the growth zones or healed microfractures [32,33]. At the Inkur and Holtoson, the lack of growth banding in quartz made it difficult to classify fluid inclusions based on the criteria above. In this study, the fluid inclusion data on quartz and fluorite were obtained from some isolated inclusions that might be primary in origin according to Roedder [34,35] or groups of inclusions that have similar vapor-to-liquid ratios, as well as heating and freezing behavior. Such fluid inclusion assemblages could provide the most reliable information [36]. Inclusions that had undergone post-entrapment processes according to [37,38] were not studied.

### 6.1. Inkur Deposit

In the main gangue mineral, quartz grains of both deposits, secondary inclusions, healing numerous cracks, tended to dominate (Figure 5a). In quartz it is very rare to find individual primary two-phase inclusions (liquid > vapor) with an average size of 10 to 25 μm (Figure 5b–f) or groups of primary inclusions (Figure 5c) located away from the healed cracks and trails of secondary inclusions. The vapor phase typically occupied 20% to 25% of the inclusion volumes.

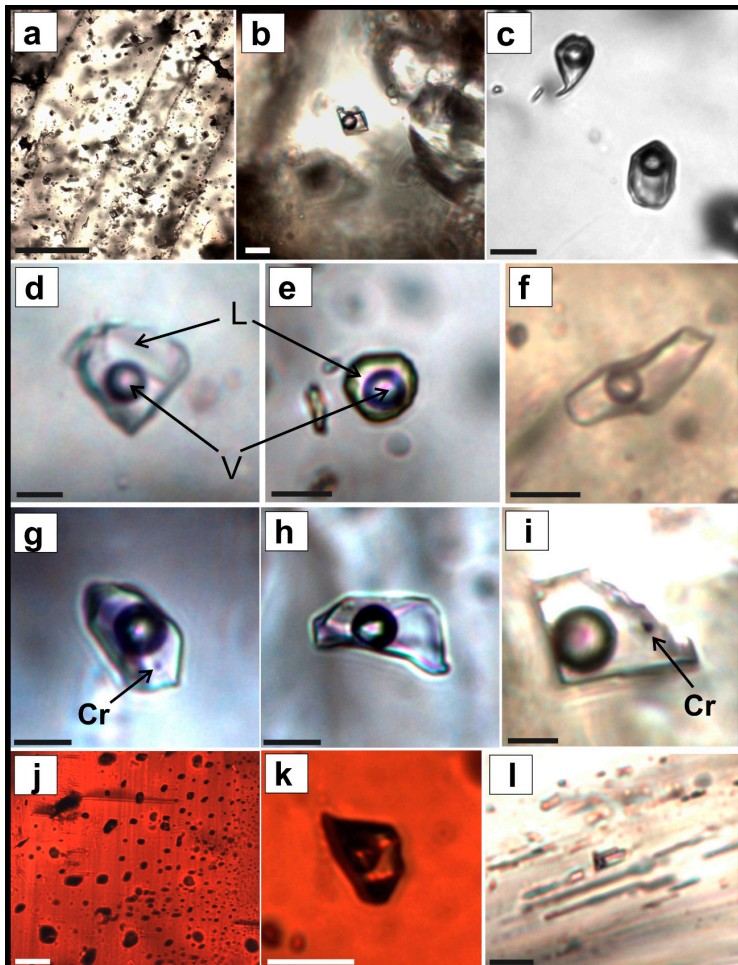

**Figure 5.** Photomicrographs of fluid inclusions from quartz-hubnerite veinlets of the Inkur deposit. (**a**) A quartz grain with trails of secondary inclusions; (**b**) primary single fluid inclusion in quartz grain; (**c**–**f**) primary two-phase fluid inclusions in quartz grains; (**g**–**i**) primary fluid inclusions with small dark phases (shown by arrows) in fluorite; (**j**) a group of vapor FIs in a hubnerite grain; (**k**) primary single two-phase FI in hubnerite; (**l**) a group of small FIs in muscovite. Cr—small dark solid phases, L—liquid, V—vapor. The scale bar length is 10 μm, for photo (**a**) 100 μm.

In rare fluorite grains, there were found primary FIs, usually single (~15–25 μm in size), which often (except for the vapor phase and the aqueous solution) contained a small solid dark phase (Figure 5g,i) of unknown composition that could not be identified by Raman spectroscopy. The vapor phase typically occupied 25% of the inclusion volumes.

In addition, in hubnerite crystals, there were observed numerous vapor inclusions (Figure 5j). In some grains, there were found very rare single two-phase (liquid > vapor) inclusions (Figure 5k) of small size (~7–12 μm). The vapor phase typically occupied 20% of the inclusion volumes.

In muscovite, there were found single groups of very small two-phase (liquid > vapor) FIs, similar to the secondary ones (Figure 5l), <10 μm in size.

Almost all inclusions belonged to FIs of homogeneous trapping; they were usually characterized by the absence of solid phases with the exception of fluorite grains. The FI thermometry and cryometry results are summarized in Table 6.

The homogenization temperature range of the studied primary inclusions in vein quartz from quartz-hubnerite veinlets varied from 343 to 195 °C (see Table 6; Figure 6). In most inclusions from quartz, the eutectic temperatures varied from ~−52 to −49.2 °C; therefore, the main salt systems were represented by $CaCl_2$-$MgCl_2$-$H_2O$, $CaCl_2$-$KCl$-$H_2O$, and $CaCl$-$H_2O$. In some inclusions, an increase in the rate of ice melting in the temperature range ~−23.4–−23 °C was observed, which may indicate the presence of $NaCl$-$KCl$-$H_2O$. The ice melting point was −4.2–−10.6 °C, respectively; the salinity varied in the range ~6.7–14.6 wt.% eq. NaCl. Here and below, the bulk salinity was calculated using the $NaCl$-$H_2O$ system according to [29].

Homogenization of the inclusions from fluorite grains occurred at relatively lower temperatures (~195–265 °C) (see Table 6). The dissolution of small solid phases was observed at temperatures of 184–187 °C.

The eutectic temperatures were close to the temperatures of FIs from quartz ~−55−49 °C with ice melting acceleration at ~−24–−23.2 °C, respectively; the main salt systems were the same: $CaCl_2$-$NaCl$-$H_2O$, $CaCl$-$H_2O$, and $NaCl$-$KCl$-$H_2O$.

The temperature ranges for homogenization of FIs hosted in hubnerite, determined by several inclusions, were narrower, 245–278 °C (see Table 6). Despite the small size, it was possible to determine the ice melting temperature (~−3.2–−3.1 °C), which corresponded to the salinity of ~5.1–5.3 wt.% eq. NaCl.

In addition, in two inclusions from muscovite, it was possible to determine the homogenization temperatures 167 and 202 °C, as well as to approximately estimate the salinity (~5.7 wt.%) eq. NaCl. Muscovite was from the central part of the veinlet; thus, most likely, it was later than the earlier muscovite from the selvage parts.

In Figure 7, there are distribution histograms of homogenization temperatures of FIs from different minerals for the Inkur and Kholtoson deposits. In the ores of the Inkur deposit, the predominant group of inclusions from quartz, fluorite, and hubnerite were distinguished with a maximum of values in the range 200–250 °C. Some of the inclusions were homogenized at higher temperatures; in this group, the modal temperature value of FIs corresponded to the range of 250–300 °C. Some FIs from quartz and fluorite were homogenized in the range 150–200 °C. According to the Raman spectroscopy data, carbon dioxide was identified in the vapor phase composition of the inclusions from the ore veinlet quartz.

**Table 6.** Microthermometry summary table of fluid inclusions in the minerals of the Inkur and Kholtoson deposits.

| Deposit | Host Mineral | FI Type | | $T_h$ | $T_{ice}$ | $T_{eut}$ | $T_{sdph}$ | Salinity Equivalent NaCl wt.% [3] | Salt System Type [4] |
|---|---|---|---|---|---|---|---|---|---|
| | | | | °C | | | | | |
| Inkur | Quartz | - | | ≥343 . . . 195 | −4.2 . . . −10.6 | −52 . . . −50 . . . −49.2 (−23.4 . . . −23—accelerated ice melting) | - | 6.7–14.6 | CaCl$_2$-MgCl$_2$-H$_2$O CaCl$_2$-KCl-H$_2$O CaCl-H$_2$O NaCl-KCl-H$_2$O |
| | Fluorite | - | | ≥265 . . . 195 | −7 . . . −3.8 | -55 . . . -49 (−24 . . . −23.2—accelerated ice melting) | 184 . . . 187 | 6.2–10.5 | CaCl$_2$-NaCl-H$_2$O CaCl-H$_2$O NaCl-KCl-H$_2$O |
| | Hubnerite | - | | ≥278 . . . 245 | −3.2 . . . −3.1 | - | - | 5.1–5.3 | - |
| | Muscovite | - | | ≥202 . . . 167 | −3.5 | - | - | 5.7 | - |
| Kholtoson | Quartz | Homogenious | | ≥344 . . . 210 | −7.2 . . . −2.9 | −38 . . . −36 −50 . . . −49 −55 | - | 4.8–10.7 | MgCl-KCl-H$_2$O NaCl-FeCl$_2$-H$_2$O FeCl$_3$-H$_2$O CaCl$_2$-KCl-H$_2$O CaCl-H$_2$O CaCl$_2$-NaCl-H$_2$O |
| | | Heterogenious | **a**—vapor-dominated | ≥413 . . . 350 (homogenization in vapor) | - | - | - | - | - |
| | | | **b**—liquid-dominated | ≥400 . . . 370 | ~−4.4 | −48 . . . 47? | - | ~7 | CaCl-H$_2$O |
| | Fluorite | - | - | ≥272 . . . 260 | −3.9 . . . −3.7 | −49 . . . −48 | - | 6-6.3 | CaCl-H$_2$O |
| | Hubnerite | - | - | ≥290 . . . 250 | −6.5 . . . −5.7 | −55 . . . 54 | - | 8.8–9.9 | CaCl$_2$-NaCl-H$_2$O |

Note: $T_h$—homogenization temperature, $T_{ice}$—ice melting temperature, $T_{eut}$—eutectic temperature, $T_{sdph}$—solid phase melting temperature, hyphen—parameter was not determined.

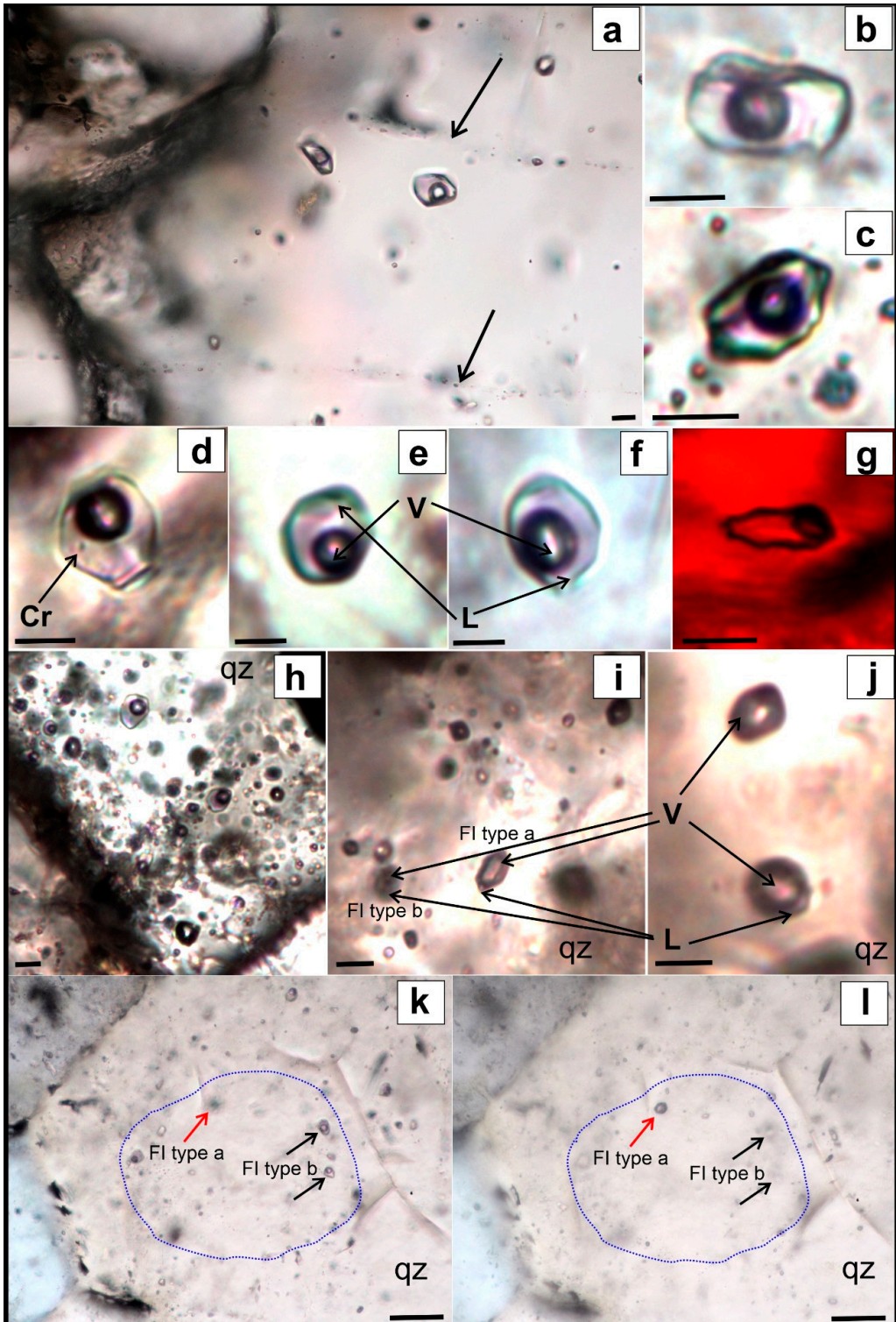

**Figure 6.** Fluid inclusions from quartz-sulfide-hubnerite veins of the Kholtoson deposit. (**a**) Quartz grain with primary two-phase fluid inclusions, which are located between healed cracks of secondary inclusions (shown by arrows); (**b,c**) primary two-phase fluid inclusions in quartz grains; (**d**) primary inclusion with small dark-colored solid phase (marked as Cr) in fluorite; (**e,f**) primary inclusions in fluorite grains; (**g**) primary two-phase FI in the hubnerite grain; (**h**) group of fluid inclusions in quartz; (**i,j**) groups of liquid- and vapor-dominated inclusions; (**k,l**) quartz grain with cogenetic liquid- (type b, black arrows) and vapor-dominated (type a, red arrow) inclusions at different levels; blue dotted line—crystal growth zone. L—liquid, V—vapor. The scale bar length is 10 μm; (**k,l**) 50 μm.

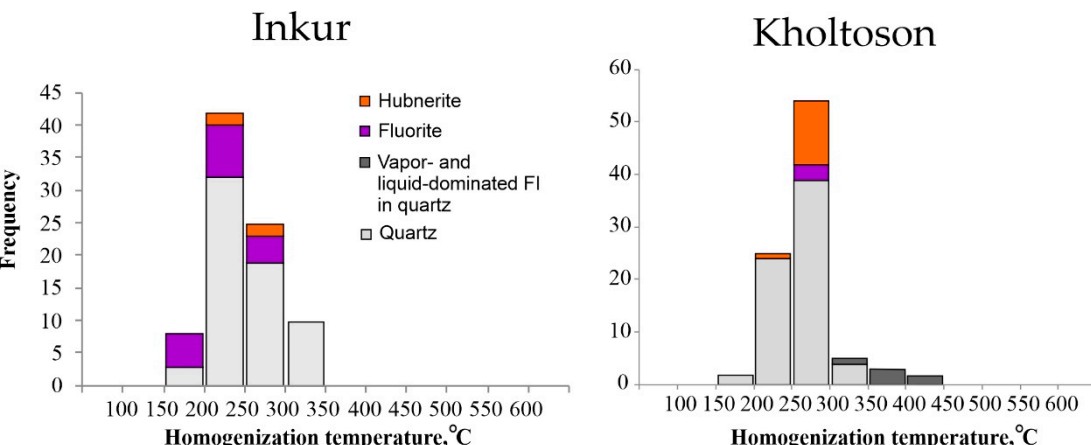

**Figure 7.** Homogenization temperature distribution histograms of fluid inclusions from different minerals of the Inkur and Kholtoson deposits.

### 6.2. Kholtoson Deposit

In quartz-sulfide-hubnerite veins of the Kholtoson deposit, the major vein mineral was represented by quartz as well; primary FIs are quite rare in the mineral. Such FIs were usually represented by two-phase inclusions (liquid > vapor), often rectangular or rounded, with an average size of 10 to 20 μm (Figure 6a–c,h). The vapor phase typically occupied 20% to 25% of the inclusion volumes.

In fluorite grains, there were also two-phase (liquid > vapor) rare primary FIs (Figure 6d–f); in some of them, there were observed very small dark-colored solid phases (Figure 6d, shown by an arrow), the composition of which could not be identified. These FIs were mostly rounded or isometric in shape, with an average size of 15–25 μm. The vapor phase typically occupied 20% to 25% of the inclusion volumes. Similarly very rare two-phase FIs of smaller sizes (≤10 microns) were also found in the hubnerite grains (Figure 6g).

Additionally, in some quartz grains, there were groups of vapor- (type a) and aqueous-dominated (type b) inclusions (Figure 6i–l), which could be conditionally attributed to inclusions of heterogeneous trapping, because they were located near each other and therefore most likely belonged to the same growth zone of the host mineral (see Figure 6k,l). The sizes of such inclusions were usually small, from 1–5 micrometers to ~8–9 μm. Most likely, the presence of such rare inclusions was evidence of episodic boiling of ore-forming solutions.

The homogenization temperature range of the studied primary FIs of homogeneous trapping varied from ≥344 to 210 °C (see Table 6; Figure 7, left) in the vein quartz. In most inclusions from quartz, the eutectic temperatures varied from ~−50 to −49 °C; in some inclusions, they were −55 °C, and less often in the range from −38 to −36 °C. Such eutectic temperatures indicated the presence of such salt components as $CaCl_2$-$KCl$-$H_2O$, $CaCl$-$H_2O$, $CaCl_2$-$NaCl$-$H_2O$, and to a lesser extent, $MgCl$-$KCl$-$H_2O$, $NaCl$-$FeCl_2$-$H_2O$, and $FeCl_3$-$H_2O$ in solutions.

Homogenization temperatures of vapor-dominated FIs (type a, see Table 6) were in the range ≥413–350 °C (Figure 7, right); the FIs homogenized to the vapor phase. The major homogenization temperatures of the FIs of homogeneous trapping from different minerals were in the range 250–300 °C (see Figure 7, right), although the bulk homogenization temperature spread of these inclusions was from 150 to 350 °C. Liquid-dominated FIs (type b, see Table 6) were homogenized in the temperature range ≥400–370 °C. Due to the small size, in only one inclusion, it was possible to approximately estimate the ice melting temperature, ~−4.4 °C, which corresponded to the total salinity of the solution of ~7 wt.% eq. NaCl.

Due to rare grains of fluorite in the quartz veins, as well as inclusions in them, it was possible to find and study several primary FIs, which were represented by inclusions of homogeneous trapping (Figure 6e,f,j), some of which had very small solid phases (Figure 6e) that could not be identified. The homogenization of such FIs occurred in the narrow temperature range $\geq$272–260 °C. The eutectic temperatures varied in the narrow range from $-49$ to $-48$ °C, which corresponded to the $CaCl_2$-$H_2O$ salt system. Ice melting temperatures were slightly lower than those of the FIs from other minerals and were within the range $-3.9$–$-3.7$ °C, which corresponded to the salinity 6–6.3 wt.% eq. NaCl (see Table 6).

The hubnerite crystals contained many vapor inclusions, and in rare grains, it was possible to find two-phase vapor-liquid FIs (Figure 6h), which are usually small in size (4–10 μm) and in some cases reach 15–20 μm. The homogenization temperatures of such FIs varied from $\geq$290 to 250 °C, and the eutectic temperatures were in the narrow range of 55–54 °C, which was closest to the $CaCl_2$-NaCl-$H_2O$ system.

The ice melting temperatures varied from $-6.5$ to $-5.7$ °C, respectively, and the salinity varied from 8.8 to 9.9 wt.% eq. NaCl.

In the vapor phase composition of the FIs from quartz, according to Raman spectroscopy data, carbon dioxide was identified.

The data on the ratios of the main salt components in FIs, obtained by the method of water extractions, indicated that in both deposits, the main salt component of solutions was Ca, whereas Na and K, and Fe and Mg, were observed in insignificant amounts. These findings were consistent with the microthermometry method data (Table 7).

**Table 7.** Content of main salt components in water extractions from FIs in quartz.

| № | Sample № | Specimen № | The Average Value of Characteristics Determined from Two Extractions, wt.% | | | | | |
|---|---|---|---|---|---|---|---|---|
| | | | Fe | Ca | Mg | Na | K | Ca/Na |
| 1 | 25−1 | Ink-11 | 0.06 | 5.43 | 0.12 | 0.84 | 0.99 | 7.45 |
| 2 | 26−1 | Ink-16 | 0.02 | 3.58 | 0.08 | 1.92 | 0.73 | 2.139 |
| 3 | 27−1 | Ink-26 | 0.03 | 2.38 | 0.05 | 2.34 | 0.63 | 1.164 |
| 4 | 28−1 | Ink-28 | 0 | 1.90 | 0.07 | 1.98 | 0.62 | 1.101 |
| 5 | 36−1 | Khol-14 | 0.02 | 5.15 | 0.19 | 0.78 | 1.37 | 7.574 |
| 6 | 37−1 | Khol-20 | 0.04 | 2.10 | 0.13 | 1.08 | 1.29 | 2.225 |
| 7 | 38−1 | Khol-23 | 0.02 | 2.28 | 0.11 | 1.25 | 0.88 | 2.092 |
| 8 | 39−1 | Khol-25 | 0.01 | 0.84 | 0.03 | 1.24 | 0.3 | 0.776 |
| 9 | 40−1 | Khol-25 | 0.34 | 0.99 | 0.16 | 1.68 | 2.16 | 0.675 |

## 7. Discussion

### 7.1. Mineralogical Characterization of the Studied Tungsten Deposits

Within the Dzhida ore field, two separate tungsten deposits, the Inkur stockwork and the vein Kholtoson, have been identified. Previously, it was considered that the formation of these deposits took place at different stages of the development of the Mo-W ore-forming system [16,20,25]. Our investigations allow us to suggest that the formation of both deposits occurred nearly simultaneously during one stage. This is evidenced by both mineralogical and fluid inclusion studies.

Hubnerite is the main ore mineral in both the Inkur and Kholtoson deposits. Based on the idiomorphism, the major part of the hubnerite was deposited during the initial period of the mineral-forming process. However, in addition to hubnerite, there are quite a large number of accompanying ore minerals present in both deposits. Mineral assemblages are similar in both deposits, which indicates similar chemistry of the ore-forming fluids. In total, more than 20 minerals were identified. There were some differences observed only in assemblages of rare and minor ore-forming minerals. The chemical composition

features of ore minerals were also similar in the ores of both deposits. For example, the muscovite, sphalerite, and tetrahedrite chemical compositions were almost identical at the Inkur and Kholtoson deposits (see Tables 2, 4 and 5). For the first time, a rare unusual mineral of the halogenide class—aluminium hydrofluoride—was found in the ores of the Inkur deposit. Its composition is similar to rosenbergite ($AlF[F_{0.5}(H_2O)_{0.5}]_4 \cdot H_2O$). In the Kholtoson deposit ores, we diagnosed unknown phases—$Cu_2PbS_3$ and $Cu_3Pb_3S_5$ as well as rare sulfosalts matildite ($AgBiS_2$) and schapbachite ($Ag_{0.4}Pb_{0.2}Bi_{0.4}S$).

The dominant gangue mineral was quartz with impurities of muscovite, potassium feldspar, and fluorite, which indicated the presence of such elements as Si, K, Al, F, and Ca in the composition of the mineral-forming fluids. The composition of ore mineral assemblages (hubnerite, sulfides, sulfosalts, etc.) indicated the presence of such elements as W, Mn, S, Fe, Cu, Pb, Zn, Bi, Be, Sn, Ag, Te, Mo, etc. in the ore-forming solutions. Thus, the solutions in addition to tungsten contained significant amounts of accompanying ore-forming elements. Nevertheless, the most economic mineral was hubnerite. This may be due either to the high W content relative to other elements in the solutions, or to specific physical and chemical parameters favorable to the deposition of hubnerite. The absence of significant differences in mineral composition of the ore veinlets of the Inkur stockwork and the veins of the Kholtoson deposit indicated the same source of the ore-forming fluids that formed the W mineralization in the Dzhida ore field.

### 7.2. Physicochemical Parameters of the Ore Formation

For both deposits, the homogenization temperatures of homogeneous trapping FIs varied in the range ~195–344 °C. These temperatures reflected minimum conditions of mineral formation [34,35]. Most FI homogenization temperature determinations at the Inkur veinlets were within the range ~200–250 °C, and at the Kholtoson veins, 250–300 °C. In both deposits, the temperature ranges of FI homogenization temperatures in hubnerite overlapped and varied within ~278–245 °C (Inkur) and ~290–250 °C (Kholtoson). According to V. B. Naumov's data [39], most of the FI homogenization temperature determinations from worldwide wolframite and scheelite are within the range 200–400 °C. The relatively medium temperature range of 265–195 °C included FIs from the Inkur deposit fluorite; slightly higher temperatures of 272–260 °C are typical for FIs from the Kholtoson deposit fluorite. Rare FIs observed in late muscovite are characterized by minimum homogenization temperatures (~202–167 °C). These inclusions are probably secondary.

The bulk salinity of the ore-forming solutions of both deposits had similar ranges (6.7–14.6 wt.% NaCl eq. for the Inkur and 4.8–10.7 wt.% NaCl eq. for the Kholtoson deposits) (see Table 2, Figure 8). These bulk salinity ranges are similar to those for many large tungsten deposits [5,6,39]. At the vapor phase, $CO_2$ was determined using Raman-spectroscopy. Absence of the liquid $CO_2$ rims in FIs suggested low $CO_2$ content in the solutions.

The main salt components of the solutions determined at both deposits were almost identical, which was confirmed by both cryometry (see Table 2) and data on water extractions (see Table 3). The main salt components were represented by calcium and sodium chlorides with minor potassium chlorides. At the same time, the fluor-containing minerals (fluorite, topaz, F-bearing muscovite, rosenbergite), represented in the ores indicated the presence of fluorine species in the fluid. According to the investigations by F. G. Reyf and E. D. Bazheev [23], the deposits of the Dzhida ore field are characterized by a fluoride admixture in the chloride-dominated hydrothermal solutions.

In the salinity–homogenization temperature diagram plots of the studied inclusions in the quartz, fluorite and hubnerite of the Inkur and the Kholtoson deposits form almost one area (Figure 8). The presence of cogenetic liquid- and vapor-dominated inclusions in the quartz from the ores of the Kholtoson deposit allowed us to estimate the true temperature range of mineral formation as 413–350 °C. The maximum homogenization temperatures were typical for the vapor inclusions that are cogenetic with the liquid ones. The presence of the rare cogenetic liquid- and vapor-dominated FIs were found in the Kholtoson veins;

it is assumed that during the ore deposition, rare periods of solutions boiling occurred. In addition, in hubnerites and sphalerites, the presence of vapor inclusions suggested that deposition probably occurred with participation of the vapor phase.

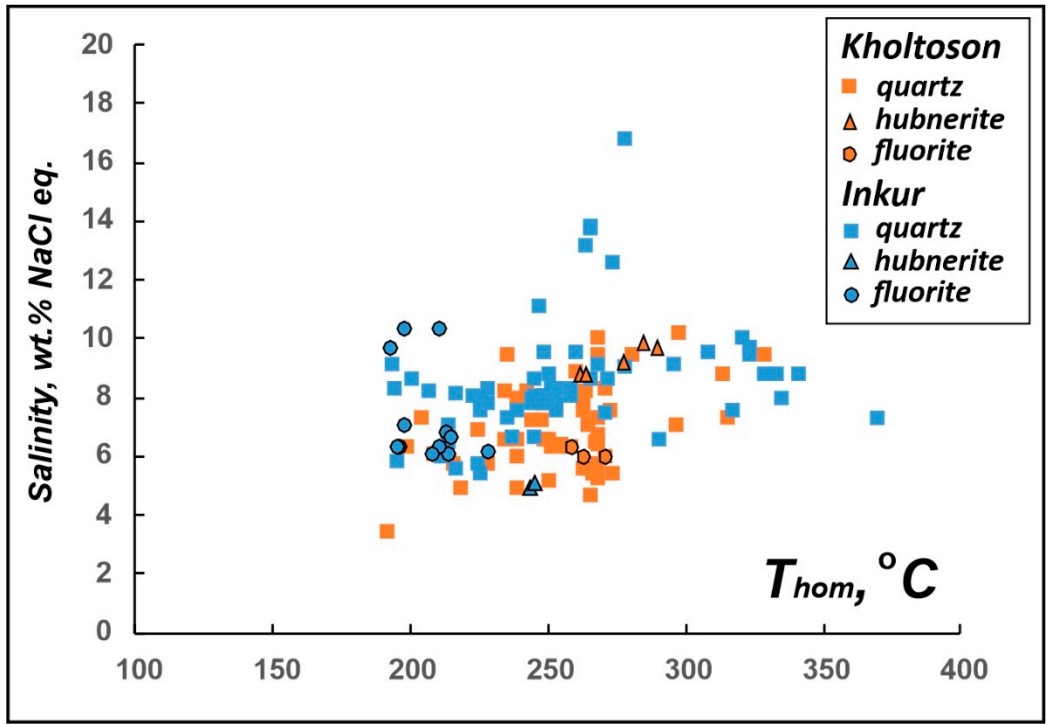

**Figure 8.** Binary salinity-homogenization temperature diagram for FIs from the Inkur and Kholtoson deposits.

### 7.3. Pressure Estimation

Assuming that the mineral formation temperatures at the Inkur and the Kholtoson were close, it is possible to calculate the pressure based on the phase relations in the NaCl-$H_2O$ system. The calculation was carried out on the difference between the maximum temperatures of homogeneous (343 °C) and heterogeneous (413 °C) trapping FIs, based on the calculation program HOKIEFLINCS_$H_2O$-NaCl, published in the article [40]. The obtained pressure value of 785 bar reflected the minimum capture pressure of FIs at the Inkur deposit. A similar value of 791 bar was obtained using the AqSo NaCl program [41]. At the Kholtoson deposit, the pressure could be estimated at about 300 bar, based on the position of the critical curve of the NaCl-$H_2O$ system according to the data [42]. The critical curve of the $CaCl_2$-$H_2O$ system had a steeper slope on the phase diagram, which indicates that the actual pressures were apparently higher than the data obtained for the NaCl-$H_2O$ system (Figure 9). The drop in pressure during mineral formation of the Kholtoson veins was probably caused by the appearance of large cracks, where quartz deposited and ore veins were formed. In contrast to the vein ores of the Kholtoson deposit, at the Inkur stockwork mineralization, signs of heterogenization were not observed, which may indicate a relatively high pressure during the ore precipitation.

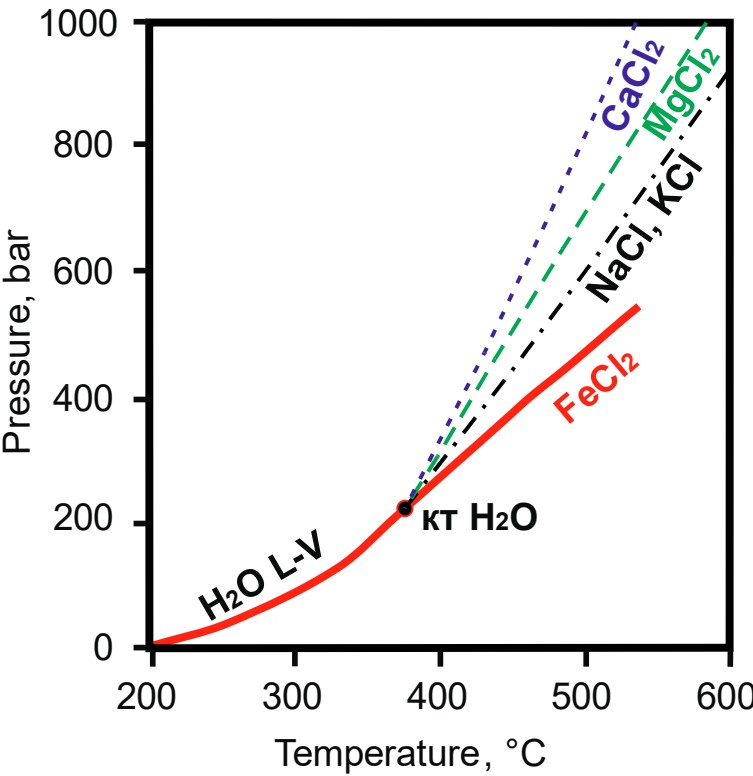

**Figure 9.** P–T projections of critical curves for water-chloride solutions [41]. L-V—liquid-vapor and KT—critical point.

*7.4. Main Factors of Hubnerite and Wolframite Precipitation*

It is suggested that the main factors of the wolframite precipitation were as follows: pressure decrease [43], cooling [44–46], wall-rock alteration [47], boiling [48,49], and fluid mixing [50,51].

Based on our study, the distribution of FI homogenization (as well as formation) temperatures in different gangue minerals of both deposits indicates that the deposition of minerals in the veinlets and veins proceeded with a decrease in temperature, whereas the pressure influence was minimal because mineral assemblages, as well as tungsten content in the stockwork and vein ores, were identical despite the pressure differences. According to the temperature changes, the bulk salinity of the solutions also decreased. This allows us to suggest that the main factors of hubnerite precipitation at the deposits of the Dzhida ore field were decreases in temperature and salinity. They could have been caused by cooling or magmatic and meteoric fluid mixing [6,9]. At the same time, wolframite solubility in the alkaline hydrothermal environment was not sensitive to the temperature or salinity of the solution, whereas in acidic solutions, a decrease in temperature leads to the precipitation of hubnerite [52,53]. The high content of muscovite in the veins and veinlets points to increased acidity of the solutions that formed the Inkur and Kholtoson deposits. According to the experimental data [52], in the acidic solutions, the solubility of hubnerite is higher than that of ferberite, which explains the W precipitation as hubnerite.

**8. Conclusions**

1. Tungsten mineralization of the Dzhida ore field is represented by the Inkur stockwork and the Kholtoson vein deposits, which were formed nearly simultaneously during one stage. Mineral assemblages are similar in both deposits, where more than 20 mineral species have been identified. The main ore mineral is hubnerite, and accompanying minerals are sulfides (pyrite, chalcopyrite, galena, sphalerite, bornite, etc.), sulfosalts (tetrahedrite, aikinite, stannite, etc.), oxides (scheelite, cassiterite), and tellurides (hessite). There were some differences observed only in assemblages of

rare and minor ore-forming minerals. For the first time, a rare unusual mineral of the halogenide class—aluminium hydrofluoride—was found in the ores of the Inkur deposit. Its composition is similar to rosenbergite ($AlF[F_{0.5}(H_2O)_{0.5}]_4 \cdot H_2O$). In the Kholtoson deposit ores, we diagnosed unknown phases—$Cu_2PbS_3$ and $Cu_3Pb_3S_5$—as well as rare sulfosalts matildite ($AgBiS_2$) and schapbachite ($Ag_{0.4}Pb_{0.2}Bi_{0.4}S$).

2. For both deposits, the fluid inclusion homogenization temperatures of homogeneous trapping FIs overlapped and varied within the range ~195–344 °C, reflecting the minimum conditions for mineral formation. The presence of cogenetic liquid- and vapor-dominated inclusions in the quartz from the ores of the Kholtoson deposit allowed us to estimate the true temperature range of mineral formation as 413–350 °C.

3. The obtained pressure value for the Inkur veinlet formation was more than 785 bar, reflecting the minimum FI trapping pressure. At the Kholtoson deposit, the pressure could be estimated at about 300 bar. The drop in pressure during mineral formation of the Kholtoson veins was probably caused by the appearance of large cracks, whereas the Inkur stockwork was formed at a relatively high pressure.

4. The deposition of minerals in the studied tungsten deposits proceeded with a decrease in temperature. The main factors of ore precipitation at the deposits of the Dzhida ore field were decreases in temperature. The acidic composition of ore-forming solutions led to the W precipitation as hubnerite.

**Author Contributions:** L.B.D. wrote the text of the article, conducted research, and provided funding. B.B.D. carried out the mineralogical study and assisted in research and sample preparation, as well as in editing and translating the text. All authors have read and agreed to the published version of the manuscript.

**Funding:** This work was supported by the Ministry of science and higher education of the Russian Federation (project GIN SB RAS No. AAAA-A21-121011390003-9). The study was performed with funding from the Russian Foundation for Basic Research, grant No. 18-45-030002p_a.

**Data Availability Statement:** The data are available upon request.

**Acknowledgments:** The authors express great thanks to analysts E.V. Khodyreva and S.V. Kanakin for conducting electron microprobe analyses (GIN SB RAS), and to N.O. Vdovenko and A.V. Sobolev for performing FI water extracts (Magadan, Russia).

**Conflicts of Interest:** The authors declare no conflict of interest.

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
