# Peer review of "Tungsten Ores of the Dzhida W-Mo Ore Field (Southwestern Transbaikalia, Russia): Mineral Composition and Physical-Chemical Conditions of Formation"

_minerals, doi:10.3390/min11070725_

Round 1

Reviewer 1 Report

Dear Editor,

I read with interest the manuscript by Damdinova and Damdinov about W deposits from the Dzhida ore district in Russia. The data are relatively well presented and illustrated, and the topic may be of interest for the community of economic geologists working on granite-related W deposits. However, the manuscript suffers from two major flaws:

  • Fluid inclusions: the description of the criteria used to distinguish primary vs secondary FIs is elusive and limited textural information is reported. Isolated or single FIs are considered as primary, but this can often be a misleading interpretation. The reported salinities are calculated in the H2O-NaCl system but the described FIs are clearly part of the H2O-CaCl2-MgCl2±KCl system as shown by the measured eutectic temperatures. Consequently, salinities need to be calculated into this system and not in the H2O-NaCl system simply. I recommend using the open-access software developed by Ronald Bakker (https://fluids.unileoben.ac.at/Computer.html) to calculate the bulk compositions of FIs. No isochore calculation is reported which do not allow to reconstruct the P-T path of FIs. The authors mention boiling as a process to explain the trapping of vapor-only FIs in some quartz grains. Caution should be addressed here for interpreting this FIA as evidence for boiling. Flashing is a common process in intrusion-related environments and can also explain the trapping of vapor-only FIs without the need for brine-vapor phase separation from a single-phase magmatic fluid.

  • Discussion: this is the major issue of the manuscript. This section mostly repeats the results about mineral paragenesis and fluid inclusions. Only five references from the literature are quoted and the data are not interpreted in regard of other granite-related W deposits in Russia or elsewhere. The P drop caused by hydrofracturing of the host rocks during emplacement of the mineralized veins is a likely mechanism for depositing hubnerite as discussed by the authors. However, because the P-T isochore trajectories are not reported for FIs, the discussion remains very general, and the model proposed is not really supported by the data.

These two major flaws make the manuscript very weak and prevents its publication in its present form. In my opinion, the manuscript requires a major revision and significant rewriting and complements added to the discussion in order to be considered for publication in Minerals. I would also suggest to the authors to find a native English speaker to proofread the revised manuscript before its acceptance. Below are some detailed comments and edits by line number to help the authors in improving their manuscript.

Best regards,

Details comments by line number

Abstract

9: remove “mineral”

18: replace “types” by “species”

21: remove “by means of”

Introduction

33: in what these deposits are “unique geological objects”?

34: replace “useful” by “economic”

41: replace “among the objects of” by “in”

54-56: the largest W deposits are located in SE China and may be referenced here

Materials and methods

64: replace “electronic” by “electron”

69: replace “the methods of thermometry, cryometry, volumometry” by “microthermometry”

70: delete “common”

72: replace “microthermocamera of the Linkam brand” by “Linkam stage”

84: are you describing here crush leach analysis of fluid inclusions?

Results

86: This section should be renamed if you only describe the local geology based on previously published studies. It could be simply renamed “Regional Geology” or “Geological setting”

90: replace “tungsten” by “W” to be consistent with the main text

98: why do you mean by “complexly constructed cover-fold area”? it is unclear to me

99: replace “faults of the Late Paleozoic stage” by “Late Paleozoic faults”

104: zones of mélange of what?

116: “porphyry granite”

121: replace “epiphyses” by “apophyses”

123: “granosyenite” is not a recognized term by the IUGS. Do you mean syenogranite? Please refer to the QAP classification diagram of Streckeisen for the nomenclature of igneous rocks

123: replace “xenoliths of roof rocks” by “roof pendants”

131: replace “at the turn of” by “about”

Deposit Geology

148: what do you mean by “presence of discontinuous 148 zones of different origin and age”?

157: replace “presence of pinches and swells of thickness” by “variable thickness”

157: delete “Industrial”. Possibly replace by “Ore”

161: “the W-bearing veinlets have…”

165: replace “crack filling” by “hydrofracturing” or “crack and seal”

172: what are “beresites”? is this a local term? If yes you should mention it

175: replace “power” by “thickness” here and in the following text

176: replace “reveal itself in the appearance of” by “consist of”

177: “fluorite”

181: “considerable distance” can you give a number in km here?

183: “granosyenite” see my comment above

189: delete “stretch and drop sizes of industrial”

189: replace “power” by “thickness”

194: “sulfosalts”

Ore Mineralogy

197: delete “Vein”. Repetitive with veinlets already

199: replace “a through” by “the main/dominant”

201: replace “connected to” by “located along”

204 : “selvageside” ???

205: replace “large-scaled” by “coarse-grained”

206: replace “impurity of F” by “high contents of F”

210: are these values determined by EDS analyses? This should be mention here and in the following text

218: “fahlore”? what is this?

219: replace “in assemblage with” by “associated with”

226: replace “in intergrowths with” by “as intergrown with”

228: replace “in assemblage with” by “associated with”

238: delete “scales”

244: what do you mean by “effusions”? do you mean “exsolution”?

247-248: mention here that these concentrations were determined by EDS analysis

252: replace “section” by “grain”

256: delete “joint”

259: replace “Hypergene secondary” by “Supergene”

271: delete “observed”

275: “alteration of”

276: a greisen is by definition an altered granite composed of secondary quartz and muscovite plus additional F-rich minerals (topaz, tourmaline, fluorite, etc). If you have muscovite selvage only, can you call this a greisenized area?

279: delete “observed”

283: replace “through” by ”dominant”

284: replace “concrescence” by “association”

287: delete “congregations or”

299: I don’t get the argument about the cubic shape of pyrite as a criterion for its relative time formation.

301: delete “solid”

302: replace “specific triangles of coloring” by “characteristic triangular polishing pits”

303: replace “concrescence” by “association”

307: delete “In”

311: replace “margins” by “rims”

322: “effusions” see my comment above

323-325: “stannite”

Fluid inclusion study

342: replace “from” by “hosted in”

344: replace “thermometry and cryometry methods” by “microthermometry”

349 and below: it is usually more precise to use the term “vapor” rather than “gas”. A vapor can be composed of one gas or a mixture of several gases.

350: replace “aprons” by “trails”

361: replace “capture” by “trapping”

368: what do you mean by “acceleration of ice melting”?

400 and 464: does the observed N2 peak was subtracted from the atmospheric N2 contribution?

402 and elsewhere: replace “inclusions” by “FIs” to be precise

442, 423, 450: “capture” see comment above

425: you mention here evidence of “boiling” but where are the brine inclusions in that case? Vapor-only FIs can also be formed by flashing or phase separation from a crystallizing pluton

468: replace “cryometric method” by “microthermometry”

Figures:

Fig 1: you should show a map of location of the Dzhida district in Russia as an inset. The North direction and GPS coordinates are missing on the map. In the caption, replace “Schematic” by “Simplified”

Fig 2: “sm”?? do you mean “cm”?

Fig 3: in the caption, replace “Interrelations” by “Petrographic characteristics”. Replace “Photos” by “Photomicrographs” and “photos made in back-scattered electrons” by “BSE images”. Mineral abbreviations should follow the official recommendations of the IMA: http://www.minsocam.org/msa/ammin/toc/abstracts/2010_abstracts/jan10_abstracts/whitney_p185_10.pdf

Fig 4: many misspelled mineral names: muscovite, scheelite, chalcopyrite, molybdenite, stannite, rhodochrosite, anglesite. In the caption, replace “mineral formation sequence” by “paragenetic sequence”

Fig 5: in the caption delete “homogeneous”

Fig 6: the caption does not match the figure

Tables:

Table 3: the concentrations for the main cations (Ca, Mg, Na, K) are reported as a few ppm. This seems problematic while you report salinities of several percent…

Reviewer 2 Report

«Tungsten Ores of the Dzhida W-Mo Ore Field (South-Western Transbaikalia, Russia): Mineral Composition and Physical-Chemical Conditions of Formation» article review

  1. The aim of the study is clarifying of the mineral composition and conditions for the formation of tungsten deposits in the Dzhida ore field. To accomplish this task, a wide range of high-precision mineralogical and petrographic studies was used, including polarization, scanning electron microscopes, thermometry methods, and others. However, it is not indicated what specific geological materials are the basis of this work, the types of samples and the sampling method are not given.
  2. In "the Results" section the geological characteristics of the studied tungsten deposits Holtoson and Inkur, genetically related to granite intrusion are given. It is emphasized that the main ore mineral in stockworks and quartz veins is the manganese variety of wolframite - hubnerite. However, the specific conditions for the formation of this mineral in ores are not considered, depending on the composition of ore-bearing fluid solutions, the role of the host environment, and other factors. The composition of Hubnerite at the micro level is not given, microinclusions and other compositional features are not indicated in comparison with the description of accompanying minerals (pyrite, chalcopyrite, sphalerite, tetrahedrite, etc.). There is also little information about another tungsten mineral - scheelite. It is advisable to provide additional materials on this issue in the manuscript, to note the features of various generations of wolframite.
  3. Section 6 provides new information on the salt composition of ore-forming solutions based on the study of fluid inclusions mainly in quartz, fluorite, and hubnerite. A large amount of factual material shows that mainly two-phase gas-liquid inclusions are observed in tungsten ores. According to the homogenization temperatures, two stages of ore formation are determined: high-temperature (>3000C) and medium-temperature, which is typical for wolframite deposits in other regions, for example, the Kalba-Narym zone of East Kazakhstan.

In general, the work was carried out at a sufficiently high scientific level. The main conclusions about the mineral composition and physicochemical conditions of ore formation of the Dzhida ore field deposits are substantiated by new factual material and make a certain contribution to the study of wolframite deposits.

The reviewer proposes to supplement the material on the sampling technique in the manuscript and to provide more detailed information on the composition of the main ore minerals - gübnerite and scheelite, if possible. It is desirable to note the practical significance of the research.

Reviewer 3 Report

This manuscript presents new data on mineralogy and fluid inclusions of two deposits in the Dzhida W-Mo ore field, at south-western Transbaikalia, Russia. However, several issues have to be addressed before this manuscript is published. They are mostly related with the re-organization of the whole manuscript, which has to be re-written in many parts. This paper mostly resembles a report to a mining company, rather than an organized paper submitted at a high-impact journal. It is characterized by numerous syntax, grammar and vocabulary errors. This causes significant problems to the reader, due to the loose structure and organization of the manuscript. Therefore, the whole text requires an extensive revision, since many parts are loose and poorly organized.

My main concerns are related to the following:

  1. Introduction must be supported by the related references which are missing.
  2. The Section 3.1. Geology of the Dzhida Ore Field must move before the Materials and Methods.
  3. A regional map, which will show the location of the Dzhida ore field and Western Transbaikalia in Russia, must either be added separately or attached in figure 1.
  4. Generally, one-sentence paragraphs are not accepted. This makes the manuscript to look like a report to a mining company.
  5. The Geology of the Dzhida Ore Field is not properly described and there are many unclear parts. For example the authors claim that the Khokhyurta suite consists of sedimentary and volcanic rocks (lines 106-107), of effusive rocks only (Figure 1), and of metamorphized (the correct is metamorphosed) sedimentary-volcanogenic rocks. The reader is confused when three different descriptions are given for one rock unit.
  6. All figures must be located in the text in a sequence, from the first to the last one. Moreover some figures (e.g. Figure 2a) are not mentioned in the text. Symbols must be added on Figure 2 because most of the descriptions at the caption can not be distinguished on the pictures, especially on figures 2c, 2e and 2f.
  7. Mineralogy must follow the typical classification of ore mineralization, which includes gangue minerals (and not vein minerals), alteration minerals, ore minerals, and supergene minerals. The term “vein minerals” involve all the minerals comprising a vein and not only the gangue minerals, and so it is erroneously used.
  8. In the section of “Mineralogy” the authors must provide in the beginning, the whole suite of minerals in each deposit classifying them in major, minor and trace minerals.
  9. Table 1 is not necessary, because this information is provided in Table 2. In Table 2 many minerals are written incorrectly (e.g. muscovite, scheelite, chalcopyrite, molybdenite, anglesite).
  10. Many terms in the text are wrong or not well defined. What is industrial vein? or low-power veinlets? or “through” mineral? or molybdenite scales? or fine-coarse-grained aggregates? or diamond shaped? They must be replaced by the correct terms.
  11. The description in the caption of figure 3 is inadequate. More details must be provided about each photo. What do the numbers 1 and 2 represent on Fig. 3d? The authors must use the abbreviations given by in Whitney and Evans (2010) Am. Mineral, 95, 183-187. It is the reference that most people use these days.
  12. Tables with microanalyses including also the chemical compositions of the ore minerals must be provided. The reader can not follow the chemical composition of the minerals, unless they are presented in tables. Are all the analyses of zincian-stannite stoichhiometric? If not, they have to be excluded.
  13. The mode of occurrence (petrography) of the fluid inclusions must be described in a most proper way. The proportions of vapor and liquid are necessary in these descriptions. The proper terms vapor and liquid must be used in the text. In addition, the criteria for classifying the primary inclusions must be provided in details. The fluid inclusions described in the manuscript are not all primary and this has to be clarified. Especially those that are included in muscovite must be evaluated with caution, because most of them are probably secondary. Post-entrapment processes possibly affected the vapor-rich fluid inclusions in hubnerite (Fig. 6 j and k). Coexistence of liquid and vapor rich inclusions is not an evidence of phase separation. They also must have similar Th, which is not the case in Kholtoson. The vapor rich inclusions possible were the result of post entrapment processes e.g. leaking. So, the interpretation of heterogeneous entrapment is wrong. In the fluid inclusiosns part and the related Discussion, the authors have to check and refer to the following publications:

Applied Sciences 2021, 11, 479, 39 p https://doi.org/10.3390/app11020479

Contrib. Mineral. Petrol. 2012, 164, 537–550 https://doi.org/10.1007/s00410-012-0749-1

Earth Planet. Sci. Lett. 2015, 417, 107–119. https://doi.org/10.1016/j.epsl.2015.02.019

Minerals 2020, 10, 182, 26 p. https://doi.org/10.3390/min10020182

Minerals, 8, 324, 26 p. (doi:10.3390/min8080324).

Ore Geology Reviews, 112 https://doi.org/10.1016/j.oregeorev.2019.103023

  1. Large part of the Discussion is the summary of the Results and must be revised. The Discussion should provide interpretations based on the literature, which are missing. The authors must provide evidence about the formation conditions of the gangue minerals, the alteration minerals and the ore minerals. Especially the endowment of the ore-forming solutions in various metals that appear in lines 488-489, based on the ore mineralogy, is not accepted. Bulk chemical analyses must be provided.
  2. Similarly, the Discussion of the fluid inclusions provide an extended summary of the results and can be omitted. The discussion of fluid inclusions must provide interpretations based on published works with similar deposits (see above). Therefore, the conclusion that the true temperature of mineral formation at Kholtoson is 350-413 °C, can not be accepted, because there is no evidence of phase separation.
  3. In the diagram of figure 8 the various FI host minerals must be indicated separately. X-axis does not show the temperature but the Homogenization temperature.
  4. The conclusion that there are two mineralization stages, one >300°C and another at <200-300°C is not supported by the results of this study, neither from the ore mineralogy nor from the fluid inclusions; unless, a more comprehensive discussion is made.
  5. Conclusions must be shortened and present only the highlights. This is a summary of the work.
  6. An important issue of this manuscript is that the English language is relatively poor. The whole text requires an extensive revision, since many parts are poorly understood.

Some specific comments to the authors:

Lines 38-39 and 41: Avoid repetitions.

Line 68: Please clarify what do you mean with the “other minerals”

The whole manuscript: it is better to use Liquid and Vapor rather than water and gas, which is not correct.

Line 88: what is a combined Mo-W mineralization?

Lines91-96: This is not the proper way to refer to previous studies. More details are required.

Line 170: What unit is sm?

Line 187: the line on the figure is re and not white.

Line 209: what is the variety wolframite-hubnerite?

Line 257: What is Pyrite II? Not mentioned before.

Lines 261-263: Why all these minerals are not shown in

Lines 273 and 280: Why K-feldspar and siderite are not presented in Table 1 and Figure 4?

Line 323: Use the term stannite instead of stannine.

Line 290: which are the minor minerals? They must be mentioned.

Lines 291 and 298: There is a conflict. The authors first write that “Hubnerite is the main ore mineral of the deposit” and then that “scheelite is more recent in comparison with hubnerite”. This must be clarified.

Line 301: All the minerals are solid. You probably mean “massive”.

Line 308-309: Cd and Fe are not impurities in sphalerite. The always are present in its chemical composition.

Line 315-317: However in Fig. 4 chalcopyrite is shown to have been formed earlier than tetrahedrite. The must be consistent.

Line 350: what do you mean by this: “groups of primary inclusions, removed from the healed cracks and aprons of secondary inclusions”. Primary inclusions do not remove from healed cracks.

Line 354: solid phases instead of dark phase.

Line 363: 343 to 195 °C – at Fig. 6, some measurements between 350 to 400 °C are also indicated.

Line 369: What are these full stops between the temperatures? The must be omitted!

Line 380: Normally when the size of the FIs is small it is not possible to determine the ice melting temperature.

Fig. 5. All the inclusions in the figure are not primary. (g-i) Please indicate the solid phases with an arrow.

Line 389: The two inclusions is muscovite are probably secondary.

Lines 398-400: Give the approximate proportions of CO2 and N.

Line 401-403: This paragraph must move before the microthermometry.

Figure 6. The description in the caption is not consistent with the description in the text. Please revise accordingly. There is not any image on Fig. 6l! The arrows in f and g, and j and k are confusing and must be omitted. What does the abbreviation Cr on Fig. 6e mean?

Figure 7. What is Syngenetic FI in quartz?

Lines 440 and 444. What are the FI-types a and b? They are not described before.

Line 456: Which is the meaning of the sentence: The hubnerite crystals are saturated with gas inclusions?

Line 475: 20 mineral types or 20 minerals?

Line 357: Boiling is not supported by the results.

Line 364: Is there sedimentation by hydrothermal solutions?

Round 2

Reviewer 1 Report

Dear Editor,

The manuscript by Damdinova and Damdinov has been significantly improved following the first revision. My main concerns on the original manuscript have been mostly fixed by the authors in their revised manuscript. I have some minor suggestions and comments (detailed below) to fix some points in the text regarding English and clarifications. My main critical comment concerns the claim of the authors for the presence of boiling during mineral deposition. This is possible, but the photomicrographs shown by the authors are not convincing for demonstrating the presence of coexisting brine and vapor fluid inclusions (boiling assemblage). I suggest here to add new photomicrographs with more convincing textural evidence of boiling. In my opinion, the manuscript requires a minor revision to be published in Minerals.

Best regards,

Detailed comments by line number:

ABSTRACT

L11: insert “spatially” between “mineralization” and “coincides”

L12: delete “the” before “numerous quartz veins”

L15: change “potassium feldspar” by “K-feldspar” to be consistent with the main text

L18: replace “determined” by “identified”

L19: replace “stannine” by “stannite”

L21 and elsewhere: replace “solutions” by “fluids”

L21-22: replace “during the existence of” by “related to”

L24: do you interpret this dilution-cooling trend as a mixing process? If yes, between what types of fluids?

INTRODUCTION

L57: delete “that form W and Mo mineralization”

L59: replace “by” by “based on”

L60-61: report the WO3 contents in metric tons of the Inkur and Kholtoson deposits

L66: replace “at” by “in”

REGIONAL GEOLOGY

L78: what do you mean by “complicated cover-fold area”? be more specific

L82: replace “submeridional” by “S-oriented”

L104: insert “spatially” between “mineralization” and “coincides”

L105: replace “of the western part” by “on the western part”

SAMPLING AND ANALYTICAL METHODS

L114: should be section 3

L118: replace “collection” by “set”

L131: replace “microthermocamera of the Linkam stage brand” by “Linkam stage”

L144: replace “for excitation” by “as excitation source”

RESULTS
L147: should be section 4. Also change the numbering for the sub-sections

L163: delete “presence of”

L186: replace “degree of metasomatic processes” by “intensity of metasomatism”

L189: replace “considerable” by “close”

L198: replace “power” by “thickness”

L239: check the Ag and Bi values, I believe there is a mistake here

L279: replace “low-power” by “thin”

L304: change “that is, it” by “which”

L305: change “potassium feldspar” by “K-feldspar” to be consistent with the main text

L311: delete “et al.”

L331: delete “a”

FLUID INCLUSION STUDY

L379: Replace “Inclusions are undergone to post-entrapment processes, which were determined according to [37,38] were not studied.” By “Inclusions that undergone post-entrapment processes according to [37,38] were not studied.”

L389: replace “water” by “aqueous”

L407-408: indicate here what system you used for salinity calculation with the associated reference

L421: replace “from” by “hosted in”

L424: same comment as above, indicate the chemical system used for salinity calculation with a reference

L450-455: these coexisting L-rich and V-rich inclusions are not evident on Figure 6i-j. I mostly see V-rich inclusions without coexisting hypersaline FIs. I suggest that you show more convincing photomicrograph of boiling assemblage if present

L457: replace “capture” by “trapping”

L483: same comment as above: detail the salinity calculation with reference

L502-504: I maintain that units shown in Table 7 are incorrect for water extractions from fluid inclusions. H2O-CaCl2-NaCl±KCl fluids should contain much more than a few ppm of these cations. Check your raw data. It is more likely that the correct unit is per cent and not ppm!

DISCUSSION

L512: replace “near-simultaneous” by “near-simultaneously”

L512-513: the existing geochronological data for the two deposits should be also mentioned here to support the simultaneous formation model.

L539: replace “at” by “in”

L582: because your FI belong to the H2O-CaCl2-MgCl2-KCl system, the isochore should be calculated in this system and not in the H2O-NaCl system. Use the Ronald Bakker’s softwares for that: https://fluids.unileoben.ac.at/Computer.html

L604: change “in” by “is”

L608: mixing with what type of fluids?

CONCLUSIONS

L618: replace “near-simultaneous” by “near-simultaneously”

L631-633: the photomicrograph that you show is not convincing to me as one cannot see the coexisting of brine and vapor fluid inclusions in the same assemblage, which would be the ultimate evidence for boiling. I suggest that you show additional photomicrograph to demonstrate textural evidence of boiling in the deposits

Reviewer 3 Report

The authors have revised the manuscript and most comments of the previous review have been successively encountered, especially at the mineralogical part. However few issues in the fluid inclusions must be addressed before this paper is published. My main concerns focus on the following parts:

  1. Lines 450-452: the classification in two types (type a and type b) must be presented here, because it is used later in the text and in Table 6.
  2. Lines 472-7473: The results of microthermometry for Type b inclusions must move here.
  3. Lines 547-548: The authors must clarify if the comment related with the Naumov’s data, refers either to the Dzhida W-Mo ore field, or to similar deposits in Russia or worldwide.
  4. Lines 566-567 and figure 8: The available Sal-Th plots of the studied inclusions in fluorite and hubernite must be added in the diagram, with different symbols, for a more complete overview of the microthermometric data.
  5. Lines 568-570: This is not shown in Fig. 8, unless the authors use different symbols or enclose these measurements within a line.
  6. Line 547: The title must change to: Main factors of hubnerite and wolframite precipitation.
  7. English requires an extensive revision.
